# Two mouse models reveal an actionable PARP1 dependence in aggressive chronic lymphocytic leukemia

Gero Knittel[1,2,3], Tim Rehkämper[1,2,3], Darya Korovkina[1,2,3], Paul Liedgens[1,2,3], Christian Fritz[1,2,3], Alessandro Torgovnick[1,2,3], Yussor Al-Baldawi[4], Mona Al-Maarri[5], Yupeng Cun[6], Oleg Fedorchenko[1,2,3], Arina Riabinska[1,2,3], Filippo Beleggia[1,2,3], Phuong-Hien Nguyen[1,2,3], F. Thomas Wunderlich[5], Monika Ortmann[7], Manuel Montesinos-Rongen[8], Eugen Tausch[9], Stephan Stilgenbauer[9], Lukas P. Frenzel[1,2,3], Marco Herling[1,2,3,10], Carmen Herling[1,3], Jasmin Bahlo[1], Michael Hallek[1,2,3], Martin Peifer[6], Reinhard Buettner[3,7], Thorsten Persigehl[4] & H. Christian Reinhardt[1,2,3,10]

Chronic lymphocytic leukemia (CLL) remains an incurable disease. Two recurrent cytogenetic aberrations, namely del(17p), affecting *TP53*, and del(11q), affecting *ATM*, are associated with resistance against genotoxic chemotherapy (del17p) and poor outcome (del11q and del17p). Both del(17p) and del(11q) are also associated with inferior outcome to the novel targeted agents, such as the BTK inhibitor ibrutinib. Thus, even in the era of targeted therapies, CLL with alterations in the ATM/p53 pathway remains a clinical challenge. Here we generated two mouse models of *Atm*- and *Trp53*-deficient CLL. These animals display a significantly earlier disease onset and reduced overall survival, compared to controls. We employed these models in conjunction with transcriptome analyses following cyclophosphamide treatment to reveal that *Atm* deficiency is associated with an exquisite and genotype-specific sensitivity against PARP inhibition. Thus, we generate two aggressive CLL models and provide a preclinical rational for the use of PARP inhibitors in *ATM*-affected human CLL.

[1] Clinic I of Internal Medicine, University Hospital of Cologne, Cologne 50931, Germany. [2] Cologne Excellence Cluster on Cellular Stress Response in Aging-Associated Diseases (CECAD), University of Cologne, Cologne 50931, Germany. [3] Center of Integrated Oncology (CIO), University Hospital of Cologne, Cologne 50931, Germany. [4] Department of Radiology, Medical Faculty, University Hospital of Cologne, Cologne 50931, Germany. [5] Max-Planck-Institute for Metabolism Research, Cologne 50931, Germany. [6] Department of Translational Genomics, University of Cologne, Cologne 50931, Germany. [7] Institute of Pathology, University Hospital of Cologne, Cologne 50931, Germany. [8] Institute of Neuropathology, University Hospital of Cologne, Cologne 50931, Germany. [9] Department of Internal Medicine III, Ulm University, Ulm 89070, Germany. [10] Center of Molecular Medicine, University of Cologne, Cologne 50931, Germany. Tim Rehkämper, Darya Korovkina and Paul Liedgens contributed equally to this work. Correspondence and requests for materials should be addressed to G.K. (email: gero.knittel@uk-koeln.de) or to H.C.R. (email: christian.reinhardt@uk-koeln.de)

C hronic lymphocytic leukemia (CLL) accounts for ~30% of all adult leukemias and 25% of non-Hodgkin lymphomas[1, 2]. It is the most common form of leukemia in the western world with an incidence rate of 4-5/100.000[2]. CLL is a disease of the elderly with a median age at diagnosis of 72 years[3]. CLL is heterogeneous in its clinical manifestation, treatment response, and course. Some patients live for decades without therapy, while others suffer from a rapidly progressing and refractory disease[4]. It is this heterogeneity, which makes treatment of CLL especially challenging.

Two common cytogenetic alterations associated with inherent resistance against genotoxic therapies are del(17p), affecting the tumor suppressor *TP53*, and del(11q), in large parts affecting the DNA damage response (DDR) kinase *ATM*, which is critical for p53 activation following genotoxic stress[5, 6]. CLL genome and exome sequencing efforts have also led to the detection of protein-damaging mutations within *TP53* and *ATM* in both treatment-naive and pretreated CLL samples[7–10]. Both del(17p) and del(11q) are also associated with poor response to the novel targeted agents, such as the Bruton's tyrosine kinase (BTK) inhibitor ibrutinib[11].

A major drawback in CLL research, and particularly in studies of high-risk CLL, is the lack of appropriate mouse models. For instance, the commonly used *Eµ:TCL1*-driven murine CLL model is typically *Atm* and *Trp53* proficient[12, 13]. Particularly B cell-specific deletion of *Trp53* and *Atm* has not been addressed in an autochthonous mouse model, thus far. When *Eµ:TCL1* mice were crossed with *Trp53*−/− mice, an aggressive CLL-like disease developed in *Eµ:TCL1;Trp53*−/− animals, which led to a substantially reduced survival, compared to *Eµ:TCL1* mice[14].

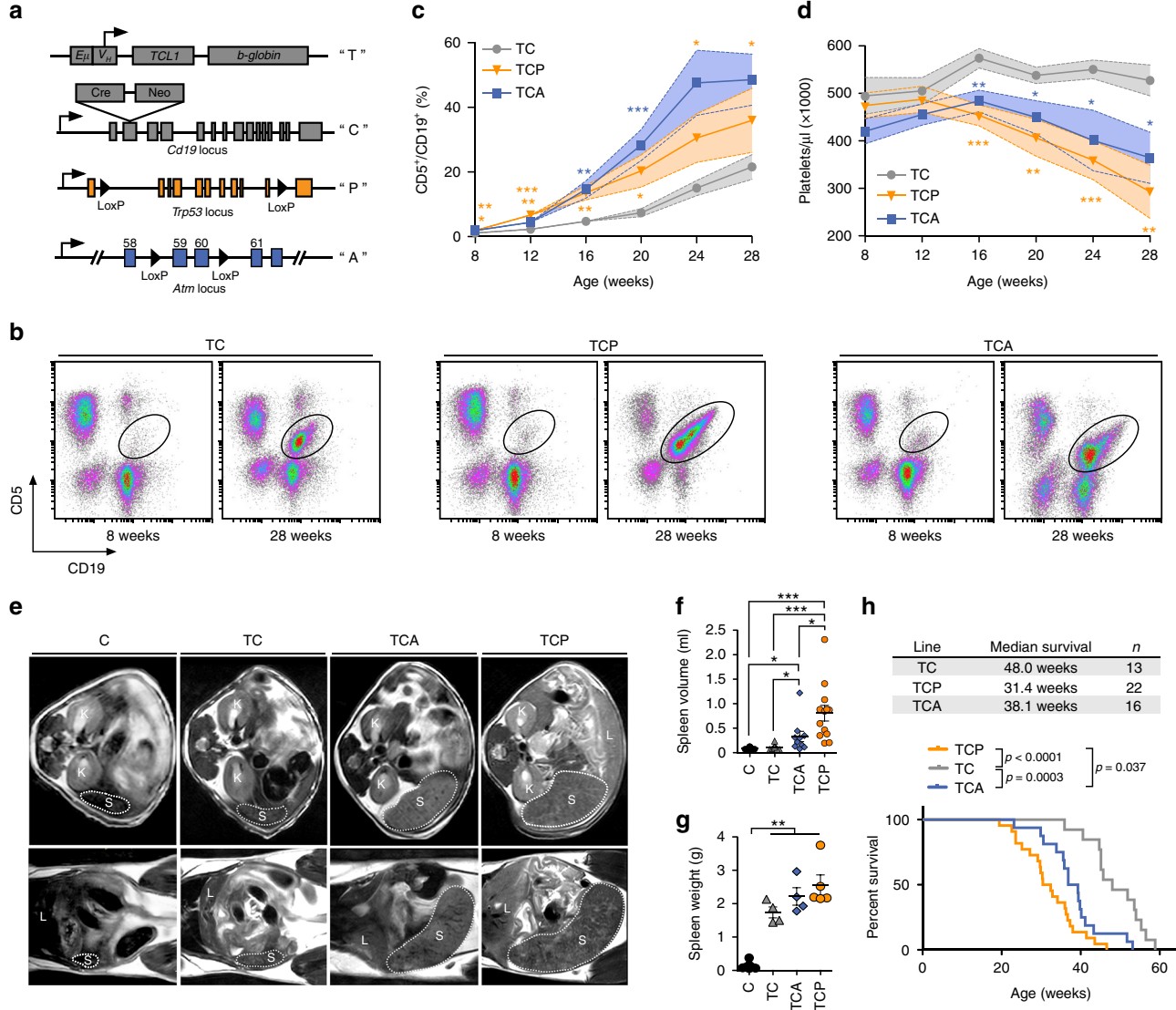

**Fig. 1** Enhanced disease progression in TCP and TCA mice. Conditional B cell-specific deletion of *Trp53* and *Atm* in *TCL1*-driven murine CLL leads to earlier disease onset and reduced overall survival. **a** Schematic illustrations of the alleles used in this study. **b** Representative CD5/CD19 plots from flow cytometric analysis of the peripheral blood of 8- and 28-week-old TC, TCP, and TCA animals. The percentage of CD5+/CD19+ cells **c** and platelet counts **d** in the peripheral blood of TC, TCP, and TCA mice ($n = 20$ for each genotype) were measured every 4 weeks, starting at 8 weeks of age. **e** Representative MR images of C, TC, TCP, and TCA mice at 30–32 weeks of age (*S* spleen, *L* liver, *K* kidney). Quantification of spleen volumes from MR images (C: $n = 11$, TC: $n = 13$, TCA: $n = 10$, TCP: $n = 13$) at 30–32 weeks of age are illustrated in **f** and spleen weights at necropsy ($n ≥ 4$/line) is illustrated in **g**. For wild-type controls, 15 animals sacrificed between 30 and 70 weeks of age were used, as a moribund state does not exist for this genotype. **h** Overall survival curves for TC, TCP, and TCA animals in Kaplan-Meier format. *Envelopes* and *error bars* represent SEM. **c**, **d**, **f**, **g** Welch's *t*-test, **h** log-rank test. *$p ≤ 0.05$; **$p ≤ 0.01$; ***$p ≤ 0.001$; ****$p ≤ 0.0001$

However, it is important to note that these *Eµ:TCL1;Trp53*$^{-/-}$ animals display altered p53 expression in the entire organism. Thus, the contribution of p53 deficiency in the CLL cells and the non-malignant stroma are impossible to dissect. Here we generated and characterized *Eµ:TCL1*-driven autochthonous models of high-risk CLL that are characterized by conditional B-cell-specific deletion of *Atm* or *Trp53*. Our novel models of *Atm-* or *Trp53*-

deficient CLL might serve as preclinical platforms for the discovery and in vivo validation of molecular liabilities associated with these high-risk genetic aberrations in CLL.

## Results

**Loss of *Atm* or *Trp53* leads to high-risk CLL in vivo**. To generate models that faithfully mimic genomic aberrations that are

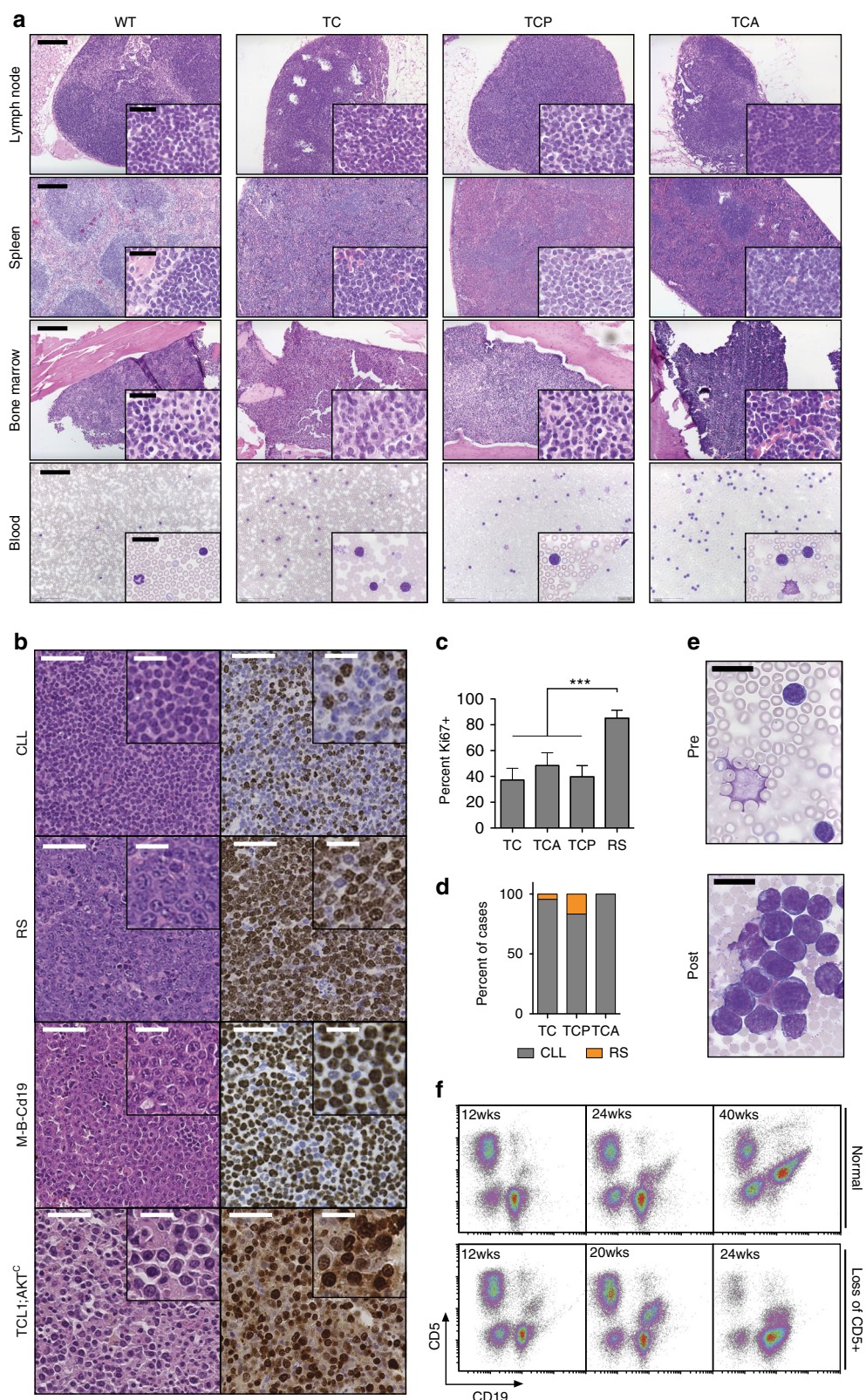

recurrently observed in high-risk human CLL, we generated animals in which B cell-specific expression of Cre recombinase leads to the conditional deletion of *Trp53* or *Atm*. We employed the *Eμ:TCL1* background and crossed in a *Cd19^Cre* allele[15], to allow B cell-specific deletion of *LoxP*-flanked *Trp53* or *Atm* alleles[16, 17] (Fig. 1a). To longitudinally monitor disease progression, we employed flow cytometry-based detection of CD5$^+$/CD19$^+$ malignant cells in the peripheral blood. Coherent with a more aggressive disease course in *Eμ:TCL1;Cd19^{Cre/wt};Trp53^{fl/fl}* (TCP) and *Eμ:TCL1;Cd19^{Cre/wt};Atm^{fl/fl}* (TCA) animals, compared to *Eμ:TCL1;Cd19^{Cre/wt}* (TC) controls, we observed a significantly higher CD5$^+$/CD19$^+$ leukemic burden in the blood of TCP and TCA animals, compared to TC mice, already at 8 weeks of age ($p$ = 0.043 for TC vs. TCP and $p$ = 0.0014 for TC vs. TCA, Welch's *t*-test, Fig. 1b, c). At 12 weeks of age, 9/20 of TCA and 12/20 TCP mice had a clearly identifiable leukemic population (CD5$^+$/CD19$^+$ ≥5%), compared to only 3/20 of TC controls ($p$ = 0.004 for TC vs. TCP and $p$ = 0.041 for TC vs. TCA, Fisher's exact test). Somewhat surprisingly, there was a trend toward an earlier occurrence of a leukemic population in TCA, compared to TCP animals (Fig. 1c). One potential mechanism for this observation may be a difference in the surface expression of homing factors between the malignant cells of the different genotypes. We addressed this possibility, by performing flow cytometry experiments to assess CXCR4, CXCR2, CD44, and CD49D surface expression on the leukemic cells. However, we did not observe any significant differences in the expression of these factors, when comparing TCA and TCP leukemia cells (Supplementary Fig. 1). Consistent with enhanced leukemogenesis and subsequent bone marrow failure, we observed significantly lower platelet counts in TCP and TCA animals, compared to TC controls, at 16 weeks of age ($p$ ≤ 0.006, Welch's *t*-test, Fig. 1d). At 24 weeks of age, 10/15 TCP and 8/13 TCA mice had developed thrombocytopenia (≤ 400,000/μl), compared to only 1/20 of TC control mice ($p$ = 0.0001 and $p$ = 0.0007, respectively, Fisher's exact test). This was paralleled by substantially enhanced leukocytosis and development of anemia in TCA and TCP animals, compared to their TC counterparts (Supplementary Fig. 2a, b). Paralleling these alterations, we also observed a substantially earlier manifestation of lymphadenopathy, assessed by magnetic resonance imaging (MRI)-based volumetry of the spleen (Fig. 1e, f). MRI scans were performed in ~30-week-old animals of all three genotypes (TC 31.1 ± 1.6 weeks, TCP 29.6 ± 0.9 weeks, and TCA 31.4 ± 1.1 weeks) and *Cd19^{Cre/wt}* control mice (C, 29.5 ± 3.3 weeks). As shown in Fig. 1f, TC mice displayed spleen volumes that are comparable to healthy C animals of the same age (98 ± 47 and 70 ± 7 μl, respectively, $p$ = 0.06, Welch's *t*-test). In contrast, spleens in TCP and TCA animals were significantly larger (806 ± 578 and 331 ± 333 μl, respectively) than those of TC and C mice (Fig. 1f). These differences were largely absent when spleen weight was assessed in TC, TCP, and TCA animals that had succumbed to their disease (Fig. 1g). Lastly, overall survival analyses revealed that TCP and TCA animals succumbed to their disease significantly earlier, than their

TC counterparts (Fig. 1h). Reminiscent of the situation in human patients[18], *Trp53* deficiency was associated with the strongest reduction in median overall survival (31.4 weeks), compared to *Atm* deficiency (38.1 weeks) and animals that develop a *TCL1*-driven CLL on an *Atm*- and *Trp53*-proficient background (48.0 weeks; TCP vs. TC, $p$ < 0.0001; TCP vs. TCA, $p$ = 0.037; TCA vs. TC, $p$ = 0.0003, log-rank test) (Fig. 1h). This effect of *Trp53* deletion was also preserved, when *Trp53* was acutely deleted in pre-existing CLLs. Specifically, 4OH-tamoxifen-mediated activation of a *Cd19^{CreERT2}* allele in leukemic *Eμ:TCL1;Cd19^{CreERT2};Trp53^{fl/fl}* animals[19] led to a marked increase in leukemic burden within 12 weeks (Supplementary Fig. 3a) and a significantly earlier CLL-associated death of these animals, compared to their *Trp53*-proficient counterparts (Supplementary Fig. 3b). We note that heterozygous *Trp53* or *Atm* deletion did not result in a significant reduction in overall survival, compared to TC animals (Supplementary Fig. 4a, b). These data indicate that the conditional B cell-specific deletion of *Atm* or *Trp53* leads to the development of aggressive CLL in vivo, reflecting the situation in human patients.

**Atm- or Trp53-deficient Eμ:TCL1-driven CLLs are oligoclonal.** We next aimed to further characterize the disease that develops in TCP and TCA mice. We initially performed histological examination of lymphatic organs (lymph node, spleen, and bone marrow) isolated from moribund TC, TCA, and TCP mice and age-matched wild-type controls and performed hematoxylin/eosin (H/E) stainings. As shown in Fig. 2a, the architecture of lymph nodes, spleen, and bone marrow was disrupted in TC, TCA, and TCP mice, compared to age-matched wild-type controls. These organs were massively infiltrated by a homogeneous population of mature-appearing lymphocytes. We note that the infiltrating lymphocytes particularly in the TCA and TCP mice did not appear to be transformed into a higher-grade lymphoma or a lymphoblastic disease (Fig. 2a, *insets*). Similar results were obtained, when we assessed blood smears from wild-type controls, TC, TCA, and TCP mice. Again, there was no gross morphological difference between CLL cells from TC and TCA or TCP animals (Fig. 2a, *bottom panels*). To determine the clonality of the splenic infiltrates that we observed in TC, TCA, and TCP mice, we next performed Southern blot analyses to detect clonal immunoglobulin (*Ig*) rearrangements. Strikingly, oligoclonal *Ig* rearrangement patterns were detected in DNA isolated from these CLL-like infiltrates in all three genotypes (TC, TCA, and TCP) (Supplementary Fig. 5a). These data strongly indicate that B cell-specific *Atm* or *Trp53* deletion in *Eμ:Tcl1*-driven murine CLL leads to the development of oligoclonal CLL. In human CLL, two subsets are distinguished: those that display unmutated *IGHV* regions, likely arising from pre-germinal center B cells and those that carry mutated *IGHV* regions, which likely indicates a post-germinal center origin. To directly ask, whether the oligoclonal CLLs that we had observed in TC, TCA, and TCP animals, underwent somatic

**Fig. 2** TCP and TCA mice develop a CLL-like disease. Conditional B cell-specific deletion of *Trp53* and *Atm* in *TCL1*-driven murine CLL leads to the development of a disease displaying morphological and histological features, as well as surface marker profiles consistent with a CLL-like disease.
**a** H/E stainings of spleens, lymph nodes, bone marrow, and blood smears of TC, TCA, and TCP mice and WT controls. Scale bars overview: 200 μm; scale bars inserts: 25 μm. **b** Representative H/E and Ki67 stainings of spleens isolated from animals with CLL-like and transformed disease (*RS* Richter's syndrome). Lymphomas of *Myd88^{condp.L252P/wt};R26^{LSL.BCL2/wt};Cd19^{Cre/wt}*[22] and *EμTCL1;Cd19^{Cre/wt};AKT^C* (Al-Maarri et al., unpublished) were included as an internal reference. Scale bars overview: 50 μm; scale bars inserts: 20 μm. **c** Quantification of the Ki67 stainings from untransformed and transformed animals ($n$ = 5 for TC, TCA, and TCP, $n$ = 3 for RS cases). **d** Quantification of the frequency of Richter transformation in the different mouse lines (TC: $n$ = 23, TCP: $n$ = 18, TCA: $n$ = 7). **e** Blood smears of a TCP animal before (age = 20 weeks) and after transformation (age = 28 weeks) show the appearance of large blastoid cells in the peripheral blood upon transformation. *Scale bars*: 20 μm. The appearance of these large cells was accompanied by the loss of CD5 expression of the leukemic clone in the peripheral blood, illustrated in **f**. *Error bars* represent standard deviation. Welch's *t*-test, \*\*\*$p$ ≤ 0.001

hypermutation, as would be expected in the case of *Eµ:TCL1*-driven murine CLL, we analyzed clonal lesions for *Igh* rearrangements by direct sequencing and detected a clonal rearrangement in all animals examined (two animals/genotype). All cases harbored a potentially functional rearrangement, except for sample #4, in which we could only detect a non-functional rearrangement, presumably derived from the other allele of the

*Igh* locus. Only the sequence derived from case #2 shows a single point mutation, which results in a mutation frequency of 0.4%. Thus, all cases are considered to belong to the unmutated subgroup of CLL (Supplementary Fig. 5b). These data indicate that CLLs developing in TC, TCA, and TCP animals are oligoclonal and arise from an *IGHV* gene unmutated precursor, as initially described for the *Eµ:TCL1* mouse[12].

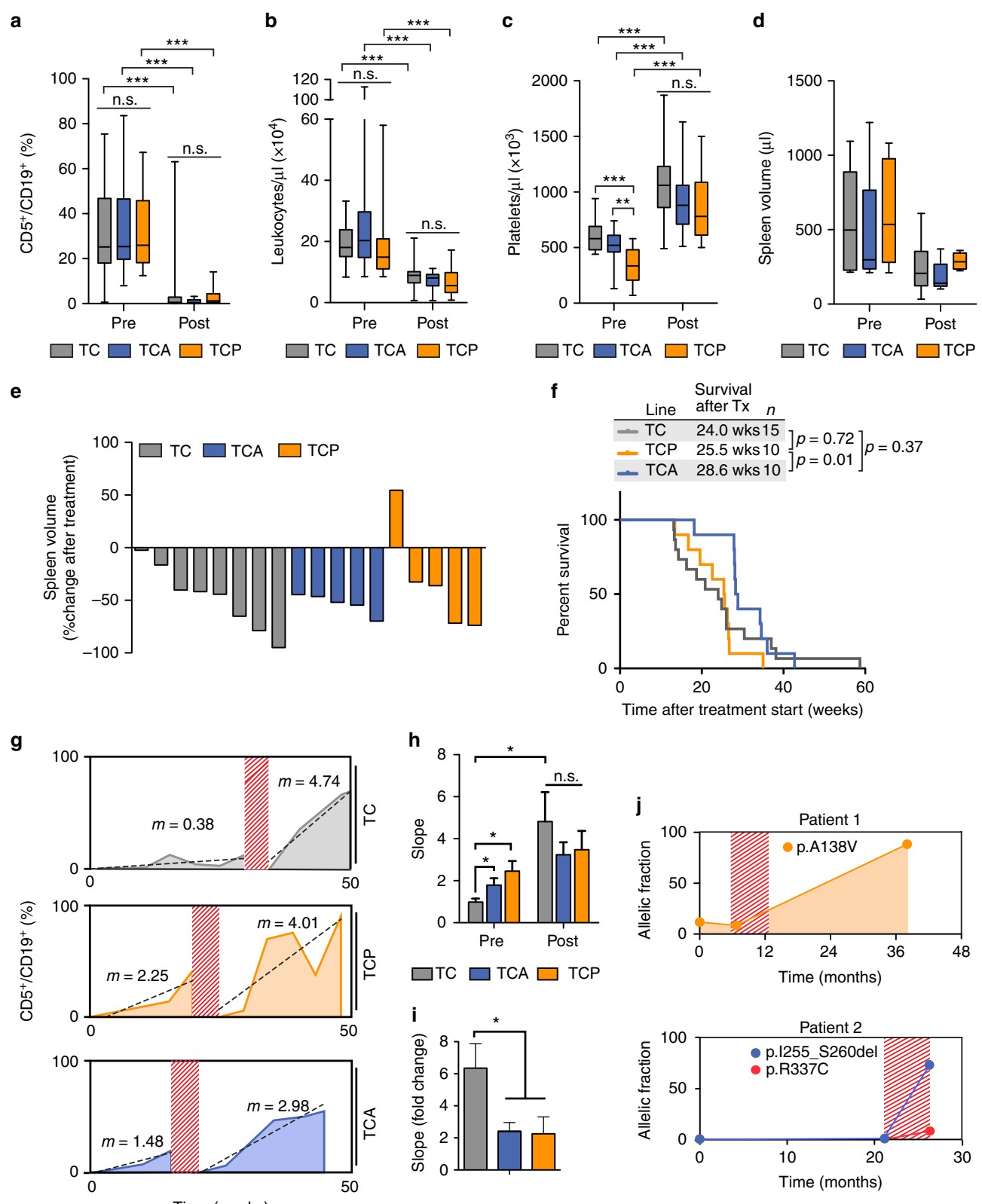

**Atm- or Trp53-deficient CLLs display occasional transformation.** A serious complication of CLL in human patients is Richter syndrome, which constitutes a rare transformation of CLL into an aggressive lymphoma type, most commonly a diffuse large B cell lymphoma (DLBCL). The molecular pathogenesis of Richter syndrome is only partially understood. Richter syndrome can occur spontaneously during CLL development, or as a form of progression and chemotherapy refractoriness. Numerous genomic aberrations have been described to be associated with Richter syndrome, including *TP53* and *NOTCH1* mutations, as well as *CDKN2A/B* deletions and *MYC* amplifications[20, 21]. Importantly, while Richter syndrome typically presents in the form of DLBCL, the genomic landscape between Richter syndrome and DLBCL appears to be largely distinct, indicating that these are indeed two different disease entities[20]. To address the question whether our novel models of high-risk CLL display an increased rate of spontaneous Richter transformation, we carefully followed cohorts of 23 TC, 7 TCA, and 18 TCP animals in a longitudinal fashion, using flow cytometry-based assessment of the leukemic clone. As shown in Fig. 2b–f, we observed occasional Richter syndrome in TC and TCP mice, but not in TCA mice. Richter transformation was characterized in these animals by the occurrence of large blastoid cells in the splenic infiltrates, reminiscent of bona fide DLBCL clones that develop in animals that co-express oncogenic *Myd88*[L252P] and *BCL2* in a B cell-specific manner[22], as well as those developing in *Eμ:TCL1;Cd19*[Cre/wt]; *Akt*[C] mice (Fig. 2b). The proliferation index, as assessed by Ki67 staining, was markedly and significantly ($p < 0.001$, Welch's *t*-test) increased in the transformed clones, compared to untransformed animals of all three genotypes (TC: $37.2 \pm 9.1\%$, TCA: $48.4 \pm 9.8\%$, TCP: $39.6 \pm 8.8\%$, transformed: $85.0 \pm 6.3\%$). Furthermore, when we analyzed the morphology observed in peripheral blood smears, we observed the appearance of large, blastoid cells in a subset of animals, in which we had also seen Richter transformation in splenic infiltrates (Fig. 2e). As loss of CD5 expression had been described in human cases of Richter syndrome[21, 23], we next performed flow cytometry-based measurements of CD5 and CD19 in CD45-gated cells isolated from a TCP animal in which we had observed the appearance of blastoid Richter-transformed cells in the peripheral blood stream. A non-transformed TCP animal served as a control. As shown in Fig. 2f, we failed to detect loss of CD5 expression in CD45$^+$/CD19$^+$ cells in the non-transformed TCP animal, while the TCP mouse with blastoid neoplastic cells in the peripheral blood displayed loss of CD5 expression on the CD45$^+$/CD19$^+$ cells. Of note, the transformed TCP animal also displayed histologically proven Richter syndrome at necropsy. Thus, overall, our novel models of high-risk CLL (TCA and TCP) do resemble the spectrum of clinical CLL cases and suffer only occasional spontaneous Richter syndrome.

**Atm- or Trp53-deficient CLLs respond to cyclophosphamide.** Next, we aimed to characterize the chemotherapy response in our novel models of high-risk CLL. For that purpose, we treated animals of all three genotypes at the time of overt CLL manifestation (defined as >20% of all CD45$^+$/SS$^{low}$ lymphocytes displaying the CD5$^+$/CD19$^+$ immunophenotype of CLL cells, in the peripheral blood) with cyclophosphamide (200 mg/kg, i.p., on days 1, 8, 15, and 22). Acute response to therapy was assessed 7 days following the last cyclophosphamide dose through flow cytometry-based quantification of the leukemic clone in the peripheral blood. As shown in Fig. 3a, b, chemotherapy treatment led to a significant reduction of the overall leukocyte count and the leukemic burden (CD5$^+$/CD19$^+$ cells) in all three genotypes. Paralleling this suppression of the leukemic cells, we observed a recovery of the platelet count in the blood stream, which we interpret as a sign of bone marrow recovery (Fig. 3c). This hematological response was also paralleled by MRI-morphological regression of splenomegaly in the cyclophosphamide-treated animals. As shown in Fig. 3d, e, MRI-based spleen volumetry demonstrated a substantial reduction in spleen volume in all three genotypes (TC, TCA, and TCP), 7 days following the last cyclophosphamide dose.

We next asked whether this initial response also resulted in prolonged overall survival in our animals. On the basis of data from human patients[18], we expected to observe a small, but significant, survival gain in TC animals, compared to TCA and TCP mice. However, the median survival gains that were induced by four cycles of cyclophosphamide did not significantly differ between the three genotypes (Fig. 3f and Supplementary Fig. 6). This surprising observation prompted us to further investigate the kinetics of the disease in TC, TCA, and TCP animals, prior and following cyclophosphamide chemotherapy in vivo. These experiments revealed that CLL progressed with significantly reduced kinetics in TC, compared to both TCA ($p = 0.041$, Welch's *t*-test) and TCP ($p = 0.015$, Welch's *t*-test) animals prior to cyclophosphamide exposure, mirroring the less aggressive disease characteristics in TC animals (Fig. 3g–i and Supplementary Fig. 7a–c). In marked contrast, leukemia progression was drastically enhanced in TC animals that had relapsed following cyclophosphamide treatment. This post-treatment progression rate was not statistically different between TC, TCA, and TCP animals ($p \geq 0.4$, Welch's *t*-test, Fig. 3h, i), strongly suggesting that four cycles of cyclophosphamide were insufficient to cure the disease in TC animals and instead selected a clone(s) that displayed similar aggressiveness as those developing in TCA and TCP animals. To further investigate the functionality of the p53 pathway in CLL cells of TC animals that had relapsed following four cycles of cyclophosphamide, we performed immunohistochemistry to assess the induction of the bona fide p53 target gene p21. These experiments revealed that exposure of

**Fig. 3** Non-curative cyclophosphamide therapy induces accelerated disease progression in TC mice. Exposure of TC animals to four cycles of cyclophosphamide-based chemotherapy is insufficient to cure the CLL that develops in these animals and instead results in a massive acceleration of disease progression, which is comparable to that observed in high-risk TCP and TCA animals. An acute response to cyclophosphamide treatment (200 mg/kg, i.p., once weekly, four injections), indicated by a decrease in the leukemic fraction (CD5$^+$/CD19$^+$ of CD45$^+$/SS$^{low}$) **a** and the total leukocyte counts **b**, as well as an increase in platelet counts **c** was observed in all three lines ($n = 14$ for each genotype, "pre", one week before treatment initiation; "post", one week after the fourth therapy cycle). **d** and **e** show spleen volumes of TC, TCA, and TCP animals ($n = 8$, $n = 5$ and $n = 5$, respectively) before and 1 week after treatment. **f** Survival after treatment initiation for TC, TCA, and TCP. **g** Percentage of CD5$^+$/CD19$^+$ lymphocytes plotted over time. Shown are representative animals of all three lines. *Dashed red areas* mark the period of cyclophosphamide treatment. The data points before and after treatment were fitted linearly, with the slope indicating the change of the leukemic fraction in the peripheral blood over time (percent points/week). **h** Averages of the change of the leukemic burden over time before and after treatment (10 animals/line). The individual time course plots are shown in Supplementary Fig. 7. **i** Fold increase of disease progression after treatment ($n = 10$ animals/line). **j** Changes in the allelic fraction over time of *TP53* mutations in two CLL patients. *Dashed areas* mark the period of therapeutic intervention (fludarabine/cyclophosphamide, one cycle (patient #1) and four cycles (patient #2)). Whiskers represent minimum to maximum, *error bars* represent SEM. **a**, **b**, **c**, **h** Welch's *t*-test for comparison between different genotypes. Paired *t*-test for comparison of pre- and post-treatment values within a cohort. **f** Log-rank test. * indicates $p < 0.05$, ** indicates $p < 0.01$, *** indicates $p < 0.001$

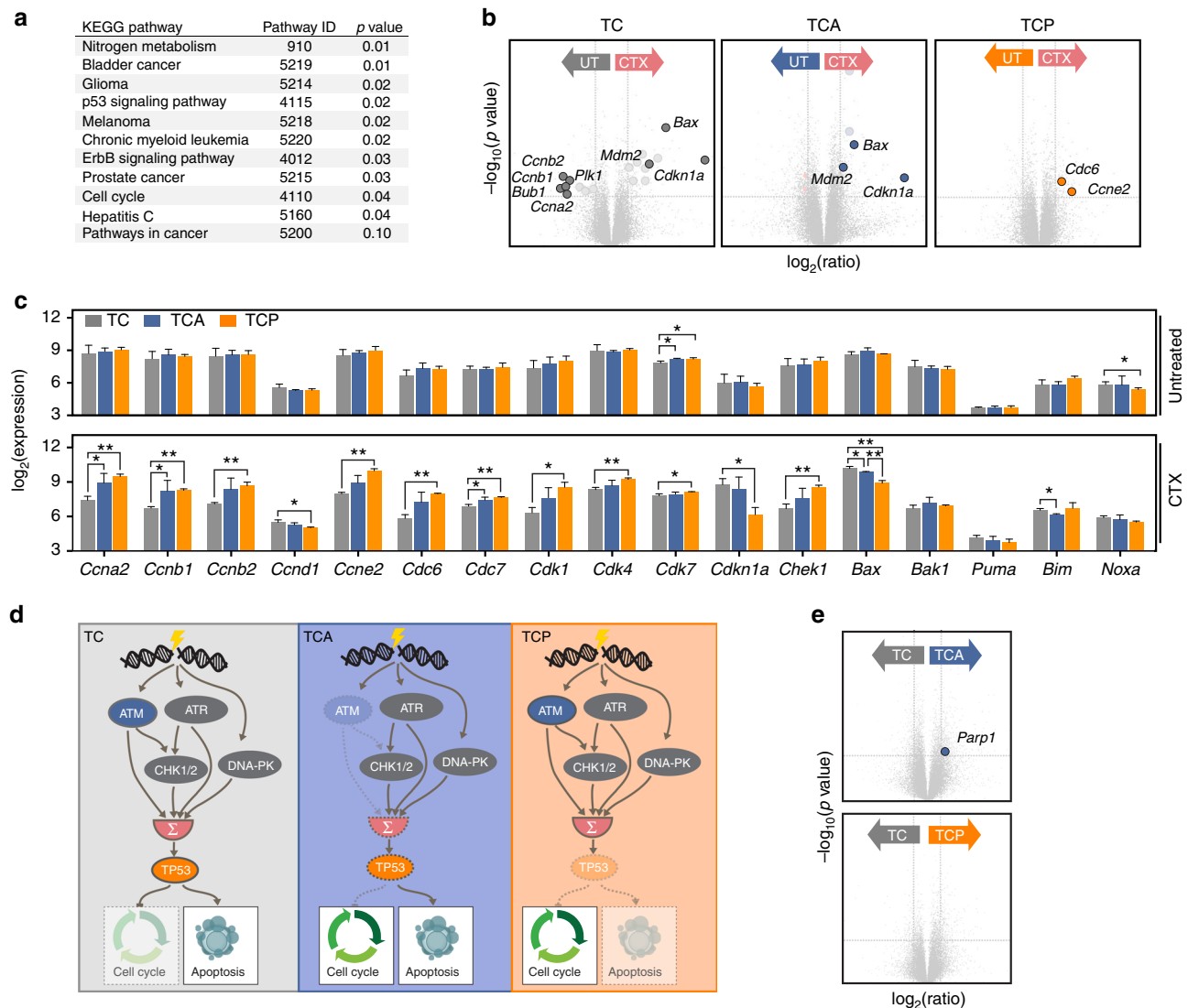

**Fig. 4** Apoptosis and cell cycle control are rewired in cyclophosphamide-treated TCA and TCP mice. Cyclophosphamide treatment induces a robust p53 response in CLL cells isolated from TC animals, evidenced by transcriptional upregulation of the p53 target genes *Mdm2*, *Bax*, and *Cdkn1a*. S-phase cyclins and mitotic genes are downregulated in cyclophosphamide-treated TC-derived CLL cells. In contrast, TCA animals lack a detectable repression of S-phase cyclins and mitotic genes in response to cyclophosphamide. Furthermore, TCP-derived CLL cells lack a transcriptional signature of a p53 response, fail to downregulate S-phase cyclins and mitotic genes and instead display an increased expression of the S-phase genes *Cdc6* and *Ccne2*. **a** Significantly differentially expressed KEGG pathways between CLL cells isolated from spleens of untreated ($n = 3$ for TC, TCA, and TCP) and cyclophosphamide-treated animals ($n = 3$ for TC, TCA, and $n = 2$ for TCP) of all three genotypes. Splenocytes were isolated 12 h after a single dose of 200 mg/kg cyclophosphamide. **b** Differentially expressed genes between untreated and treated samples of each genotype. Significantly differentially expressed components of the KEGG pathways "p53 signaling pathway" and "cell cycle" are *highlighted* (*transparent circles*) and a selected set of these genes was annotated (*colored circles*). A complete list of the significantly and differentially expressed genes is provided in Supplementary Table 1. **c** Expression levels of cell cycle and apoptosis genes in splenocytes of untreated and cyclophosphamide-treated animals (genotype-stratified). **d** Simplified illustration of the cellular response to genotoxic stress observed in leukemic clones that develop in TC, TCA, a`nd TCP animals. Integrated signals from multiple pathways activated by DNA damage lead to the activation of p53, which mediates cell cycle arrest and apoptosis. This p53-mediated response is partially impaired in *Atm*-deficient cells and completely abolished in the *p53*-deficient setting. **e** Significantly differentially expressed components of the base excision repair (*BER*) pathway between the different genotypes are *highlighted*. A complete list of differentially expressed genes of the KEGG pathway "base excision repair" is provided in Supplementary Table 2. *Error bars* represent standard deviation. Student's *t*-test, *$p \leq 0.05$, **$p \leq 0.01$

chemotherapy-naive leukemic TC animals led to a substantial induction of p21 in the spleen-infiltrating leukemic cells (Supplementary Fig. 8a, b). In marked contrast, p21 was only marginally induced in spleen-infiltrating CLL cells of TC animals that were re-challenged with cyclophosphamide, following relapse after four cycles of cyclophosphamide (Supplementary Fig. 8a, b). In line with these data, we found that p53 Ser-18 phosphorylation

was strongly induced in spleen lysates of leukemic TC animals upon cyclophosphamide treatment (Supplementary Fig. 8c). In contrast, p53 Ser-18 phosphorylation was induced by cyclophosphamide re-challenge only in ~50% of spleen lysates derived from TC animals that had relapsed following four cycles of cyclophosphamide (Supplementary Fig. 8d). Lastly, we assessed the mutation status of *Trp53* in CLL-infiltrated spleens of

chemotherapy-naive TC mice and in TC animals that had relapsed after four cycles of cyclophosphamide. For that purpose, we designed primers to amplify the regions coding for the DNA-binding domain, which harbors most mutation hot spots found in human cancers[24]. However, after deep sequencing of the amplicons at a minimum coverage of more than 20,000× (Supplementary Table 1), we did not detect any mutation in either the primary or the relapsed CLL. Together, these observations suggest that the p53 pathway is altered in relapsed animals through mechanisms that are not dominated by protein-damaging *Trp53* mutations.

These in vivo observations are corroborated by clinical data from our patient database. Specifically, we identified two individual patients that received fludarabine/cyclophosphamide-based chemotherapy and monitored the occurrence and clonal dynamics of CLL-associated high-risk mutations, using a targeted next-generation sequencing approach. Both patients displayed p53 protein-damaging mutations (p.A138V (patient #1) and pI255-S260del and p.R337C (patient #2)), which were absent (p.R337C) or detectable at very low allelic frequencies prior to fludarabine/cyclophosphamide chemotherapy (one cycle for patient #1 and four cycles for patient #2). Following chemotherapy, the allelic fraction of these high-risk genomic aberrations increased substantially (Fig. 3j), strongly suggesting that aggressive clones can be selected under genotoxic chemotherapy.

To further characterize the chemotherapy response in the TC, TCA, and TCP models, we next performed microarray-based gene expression analysis on splenocytes derived from leukemic animals with MRI-morphologically verified splenomegaly. At the time of sacrifice, TC animals were 42.8 ± 5.4, TCA animals were 31.8 ± 7.5, and TCP mice were 34.6 ± 5.2 weeks of age. In unsupervised hierarchical clustering, no obvious genotype-directed clustering of the individual models was observed (data not shown). However, we could detect substantial differences in gene expression between untreated and treated animals (all three genotypes were pooled for this analysis) (Fig. 4a). To gain further inside into the differences between treated and untreated samples, we next examined which signaling pathways were most differentially regulated between these groups, using a KEGG pathway-based analysis[25]. These experiments revealed that the KEGG terms "Trp53 signaling pathway" and "cell cycle" were significantly differentially expressed between treated and untreated animals ($p = 0.02$ and $p = 0.04$, respectively, GOstats[26]) (Fig. 4a). A genotype-stratified examination of the expression levels of the components of the kyoto encyclopedia of genes and genomes (KEGG) pathways "cell cycle" and "p53 signaling pathway" (KEGG entries mmu04110 and mmu04115, respectively) revealed a significant induction of the p53 target genes *Bax*, *Cdkn1a*, and *Mdm2* in TC animals, while S- and G2-phase cyclins (*Ccna2*, *Ccnb1*, and *Cnb2*), as well as mitotic genes (*Bub1* and *Plk1*) were significantly underrepresented (Fig. 4b and Supplementary Table 2). Furthermore, TCA animals displayed signs of a functional p53 response, with induction of *Bax*, *Cdkn1a*, and *Mdm2* in response to cyclophosphamide treatment (Fig. 4b and Supplementary Table 2). However, the therapy-induced repression of cell cycle-driving genes that we had observed in TC animals was not detectable in TCA mice (Fig. 4b and Supplementary Table 2). Lastly, TCP animals lacked the induction of p53-regulated genes in response to cyclophosphamide treatment, whereas they surprisingly displayed a significant upregulation of the G1/S cyclin *Ccne2* and *Cdc6*, which is involved in DNA replication (Fig. 4b and Supplementary Table 2). These observations suggest a model in which apoptosis-promoting genes are upregulated and genes that drive cell cycle progression are downregulated in a *Trp53*-dependent manner (Fig. 4b, c and Supplementary Table 2), following

cyclophosphamide exposure. In the *Atm*-deficient setting, the ability of CLL cells to induce cell cycle arrest appears to be impaired, due to the lack of this important DNA damage-sensing protein (Fig. 4b, c and Supplementary Table 2). However, activation of *Trp53*-mediated apoptotic signaling still occurs, most likely via other proximal DDR kinases sharing substrate homology with ATM, such as ATR or DNA-PKcs (illustrated in Fig. 4d). In line with this hypothesis, we found that CHK1 is phosphorylated on the ATR substrate site Ser345-Gln346 in lysates generated from leukemia-infiltrated spleens isolated from TCA animals 12 h following cyclophosphamide challenge, in vivo (Supplementary Fig. 9). The Ser-345 residue has previously been shown to be phosphorylated in an ATR-dependent manner[6, 27–29]. Moreover, we also observed robust phosphorylation of Ser-18 in p53 in these lysates (Supplementary Fig. 9). The Ser-18 site is a well-established substrate of the Ser-Gln-directed proximal DDR kinases ATM, ATR, and DNA-PKcs[6, 24]. Thus, overall, these immunoblot experiments strongly indicate that ATR and/or DNA-PKcs remain active in TCA CLL cells.

One of the most striking observations within our transcriptome analyses was the upregulation of the G1/S cyclin gene *Ccne* in TCP animals following cyclophosphamide (Fig. 4b). These data could suggest that CDK4/6 inhibition may be particularly effective in TCP animals. To directly address this possibility, we performed a set of in vivo experiments to assess a potential therapeutic efficacy of the CDK4/6 inhibitor palbociclib in TC and TCP animals. These experiments revealed that palbociclib did induce a mild reduction in splenomegaly in both leukemic TC and TCP animals (Supplementary Fig. 10a, b). However, these effects did not reach statistical significance and were observed both in TC and TCP animals, arguing against a genotype-specific effect (Supplementary Fig. 10a, b). In addition, we did observe a mild palbociclib-induced reduction in leukemic cells in the peripheral blood of TC and TCP animals (Supplementary Fig. 10c). However, the differences between the two genotypes were not significant. Overall, these data indicate that palbociclib displays a mild, but not TCP-specific, anti-leukemic effect in our CLL mouse models.

**Atm-deficient CLLs display an actionable addiction to PARP.** Disabling *ATM* mutations or del(11q) and particularly bi-allelic loss of *ATM* are high-risk aberrations in human CLL[18, 30, 31]. Furthermore, our data indicate that TCA mice resemble this clinical scenario (Fig. 1 and Supplementary Fig. 2) and might thus serve as ideal preclinical tools to assess the efficacy of molecularly targeted therapeutic intervention strategies. In addition to mediating apoptosis via p53 activation, ATM is also involved in DNA double-strand break (DSB) repair[32]. Mammalian cells employ two distinct pathways for DSB repair—homologous recombination (HR) and non-homologous end joining (NHEJ)[32]. Upon initiation of the HR process, the DSB is resected leading to the generation of a single-stranded (ss) 3′-overhang, which is rapidly coated by RPA[32]. Once this RPA-coated ssDNA filament is generated, RPA is replaced by Rad51 in an ATM/CHK2/BRCA1/BRCA2/PALB2-dependent fashion[32]. Rad51 ultimately mediates homology search, strand exchange, and Holliday junction formation[32]. Intriguingly, a defective HR mechanism in *BRCA1*- or *BRCA2*-deficient settings had previously been shown to be associated with an actionable dependence on Poly (ADP-ribose) polymerase 1 (PARP1)[33–35]. Given the role of ATM in the HR process and extrapolating the PARP1 dependence of *BRCA1*- or *BRCA2*-deficient cells and tumors, we speculated that our *Atm*-deficient CLL models might display a similar PARP dependence. We note that our transcriptome analysis of cyclophosphamide-treated animals revealed that *Parp1* is

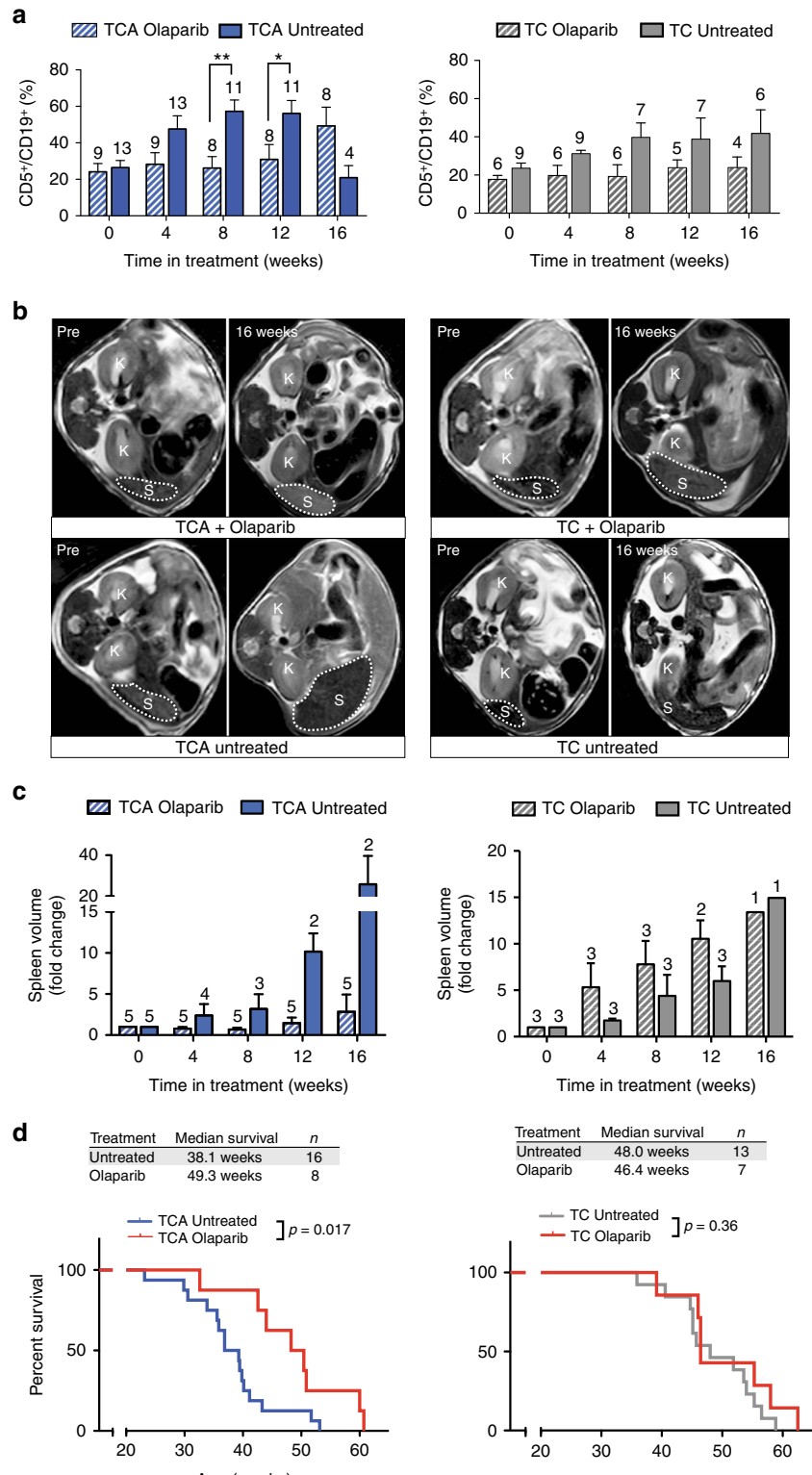

**Fig. 5** *Atm*-deficient CLLs display an actionable dependence on PARP. ATM has been shown to regulate HR-mediated DNA DSB repair. To directly address a potential dependence of *Atm*-deficient CLLs on the PARP-dependent BER pathway, we treated leukemic TC and TCA animals with the PARP inhibitor olaparib (50 mg/kg, i.p., daily, 5 days/week) and monitored disease progression and overall survival. Control cohorts were left untreated. The $CD5^+/CD19^+$ fractions of peripheral blood lymphocytes of olaparib-treated or untreated TCA and TC animals are illustrated in **a**. *Numbers* above bars indicate the number of mice that were alive at the time of imaging in each group. Lymphadenopathy was monitored through MR imaging in 8 week intervals. Representative images of mice scanned at day 0 and day 112 of treatment/observation are shown in **b**. The fold change of spleen volumes of olaparib-treated and untreated mice is illustrated in **c**. **d** Overall survival of TC and TCA mice treated with olaparib or left untreated. *Error bars* represent standard deviation. **b**, **c** Welch's *t*-test, **d** log-rank test. *$p < 0.05$, **$p < 0.01$

significantly overexpressed in TCA animals, compared to their TC counterparts, further suggesting that PARP activity might be critical for the survival of *Atm*-defective CLLs (Fig. 4e and Supplementary Tables 2, 3). We thus exposed leukemic (defined as ≥15% of all CD45$^+$/SS$^{low}$ cells displaying the CD5$^+$/CD19$^+$ immunophenotype of CLL cells and/or having a spleen volume of ≥200 μl) TC (median age of 31.8 ± 4.0 weeks at treatment initiation) and TCA mice (median age of 23.6 ± 6.1 weeks at treatment initiation) to olaparib (50 mg/kg, intraperitoneally (i.p.) on 5 days/week) and assessed leukemia progression through flow cytometry-based quantification of CD5$^+$/CD19$^+$ leukemic cells in the blood stream and MRI-based quantification of spleen volume, as a surrogate marker for lymphadenopathy. As shown in Fig. 5a–c, olaparib treatment of TC animals did not result in any reduction in leukemic cells or spleen volumes. In marked contrast, the percentage of leukemic cells was stable in olaparib-treated TCA animals up to 16 weeks following initiation of treatment, while it was increasing in untreated TCA animals (Fig. 5a–c). This difference between olaparib and untreated TCA animals was statistically significant at 2, 4, and 8 weeks of treatment. We note that the cohort of untreated TCA animals shrank from initially 15 to 2 animals at 16 weeks, due to CLL-mediated death of these control animals. Spleen volume assessment showed similar results as the assessment of the percentage of leukemic cells. As shown in Fig. 5c, the spleens of olaparib-treated TCA mice displayed a volume reduction at weeks 4 and 8 and only mildly progressed at weeks 12 and 16 under continued olaparib exposure. In marked contrast, the spleen volume of untreated TCA mice massively increased, particularly at week 12 (10.2 ± 2.2-fold compared to 1.5 ± 0.7-fold in olaparib-treated TCA mice) and week 16 (25.8 ± 13.8-fold compared to 2.8 ± 2.1 in olaparib-treated TCA mice). In perfect agreement with these data, we observed a statistically significant overall survival gain in olaparib-treated TCA mice, compared to untreated controls (median overall survival 49.3 weeks, compared to 38.1 weeks, $p = 0.017$, log-rank test) (Fig. 5d). Contrary to the situation in TCA mice, TC mice did not benefit from olaparib treatment and no statistically significant differences in overall survival were observed in these cohorts (Fig. 5d). Altogether, these data clearly indicate that *Atm*-deficient CLL displays an actionable dependence on PARP and implicate PARP inhibitors as a potential further modality in the treatment armamentarium for *ATM*-deficient high-risk human CLL.

## Discussion

CLL is the most common leukemia in the western world and remains a largely incurable disease[2, 36]. Genome sequencing recently revealed a fine-grained genomic understanding of CLL[7–10]. These biological insights fundamentally changed our approach to CLL therapy[2, 36]. Traditionally, patients with primarily indolent disease were included into surveillance programs and therapy was withheld until patients became symptomatic[2, 36]. Patients with active, progressive disease traditionally received combination chemo-immune therapy[36]. Some patients with poor prognostic features, such as del(17p) or bi-allelic *TP53* mutation, were treated with an upfront allogeneic stem cell transplantation in curative intent[37]. Although these algorithms are largely still valid today, we have witnessed paradigmatic changes in our approach to CLL treatment during the past years. The introduction of novel targeted agents into our therapeutic armamentarium, such as the BTK inhibitor ibrutinib, the PI3Kδ inhibitor idelalisib, the BCL2 inhibitor venetoclax, or the novel humanized and glycol-engineered CD20 antibody obinutuzumab, opened new therapeutic horizons[2]. These novel agents are currently evaluated as first-line regimens, both as single agents, as

well as components of combination regimens. It is anticipated that these novel drugs will eventually replace current standard chemo-immune therapies as first-line options in CLL[38]. Despite the efficacy of these new agents, there is currently no curative therapy for CLL patients besides allogeneic stem cell transplantation, for which most patients do not qualify due to age and lack of fitness[39–41]. Most importantly, it is known that high-risk genetic aberrations, particularly del(17p) and *TP53* mutations, retain their high-risk profile, even in response to venetoclax, ibrutinib, and idelalisib[11, 38, 42]. We generated models of high-risk CLL through B cell-specific conditional deletion of *Trp53* or *Atm* in *Eµ:TCL1*-driven CLL. These models mimic the human disease. Both TCP and TCA mice develop a CD5$^+$/CD19$^+$ CLL, which rarely undergoes Richter transformation (Figs. 1 and 2). Most importantly, both TCP and TCA animals succumb to their disease significantly earlier, than their standard-risk TC counterparts (Fig. 1h). Our data further indicate that heterozygous deletion of *Trp53* or *Atm* does not per se lead to a significant reduction in overall survival, compared to the *Eµ:TCL1;Cd19$^{Cre/wt}$* parental model (Supplementary Fig. 4). However, we note that we have not challenged the heterozygotes with genotoxic therapy. These experiments may reveal important mechanisms of chemotherapy-induced selection of aggressive clones, which could be mediated by loss of heterozygosity or the acquisition of protein-damaging mutations on the remaining *Trp53* and *Atm* alleles. Mechanistically, we showed that loss of p53 in the *Eµ:TCL1* background leads to a significantly reduced expression of p53 target genes, such as *Bax* and *Cdkn1a* (Fig. 4c), in response to genotoxic chemotherapy. Furthermore, the gene expression profiles obtained from leukemic TCA animals in response to cyclophosphamide suggest that the expression of pro-apoptotic p53 target genes remains intact in these CLLs (Fig. 4b). However, TCA animals display a relative overexpression of S-phase and mitosis promoting genes, such as *Ccna2*, *Ccnb1*, and *Bub1* (Fig. 4c). These data might suggest a failure to properly arrest the cell cycle in response to genotoxic damage.

In addition to regulating cell cycle checkpoints and apoptosis, ATM has also been shown to be involved in DNA DSB repair[6]. There is strong evidence for a role of ATM in HR-mediated DSB repair, with less pronounced effects on NHEJ[32, 43]. Cells derived from *Ataxia telangiectasia* (A-T) patients show a subtle, but distinct defect in DSB repair, which is due to impaired assembly and functioning of RAD51-associated protein complexes in the HR arm of DSB repair[44–46]. Recruitment of RAD51 to DSBs requires resection of DNA ends to generate RPA-coated 3′-ss overhangs. This resection process and hence the RAD51 focus formation has been shown to be ATM dependent[32, 43]. ATM has recently been shown to be required for the HR-dependent DSB repair component in G$_2$. This notion is supported by the observation that ionizing radiation-induced sister chromatid exchanges in G$_2$ require ATM[47–49]. Lastly, ATM appears to specifically mediate HR-dependent DSB repair in heterochromatin (HC). Indeed, ATM directly phosphorylates the HC-building factor KAP-1. This KAP-1 phosphorylation is critical to allow HR-mediated repair in HC areas and KAP-1 depletion is able to rescue the ATM-dependent repair defect in G$_1$ and G$_2$[49–51]. Thus, the apoptosis-evading effect of ATM deficiency likely comes at the cost of a reduced ability to repair chemotherapy-induced DSBs via HR.

HR defects, particularly in *BRCA1*- and *BRCA2*-deficient cells and tumors, have been shown to lead to PARP1 inhibitor sensitivity[33–35]. In agreement with an actionable HR defect, numerous pieces of circumstantial evidence suggest that PARP1 inhibition might be a viable therapeutic strategy in *ATM*-defective settings. For instance, RNA interference-mediated *ATM* depletion was shown to lead to olaparib sensitivity in MCF7

and ZR-75-1 breast cancer cell lines, in vitro[52]. Furthermore, in gastric cancer, low levels of ATM expression are associated with increased sensitivity to olaparib/paclitaxel combination therapy[53]. Moreover, olaparib displayed activity in ATM-defective lymphoblastoid cell lines, the *ATM*-mutant mantle cell lymphoma cell line Granta-519, and *ATM*-defective primary CLL cells[54, 55]. However, the *ATM* status-dependent specificity of PARP inhibitor sensitivity recently came into question, when it was shown that the PARP1/2 inhibitor talazoparib (BMN-673) displayed activity against CLL cells ex vivo independent of *ATM* and HR status[56]. Here we developed an autochthonous, non-transplant mouse model of *ATM*-deficient CLL to unequivocally demonstrate the in vivo activity and specificity of olaparib against *ATM*-defective CLL. Our data thus implicate olaparib as a feasible therapeutic agent in the treatment of *ATM*-defective high-risk CLL.

## Methods

**Experimental animals**. B cell-specific loss of *Atm* and *Trp53* in *Eμ:TCL1* mice[12] was modeled by crossing the *Eμ:TCL1* allele with *Cd19^Cre*[15] and either *LoxP*-flanked *Atm*[17] or *Trp53*[16] alleles on a mixed C57BL/6J-C57BL/6N background. For the tamoxifen-inducible activation of Cre recombinase, the *Cd19^CreERT2* allele[19] was used in a hemizygous state on a mixed *C57BL/6J-C57BL/6N* background.

For overall survival analysis and cyclophosphamide treatment experiments, animals were randomly assigned to the cohorts prior to the start of observation of disease progression. Treatment was initiated, when the treatment criteria were reached. For olaparib and palbociclib treatments, mice were distributed between the treatment and the observation group, in order to provide a balanced distribution of sex, pretreatment spleen volume, and age. The only exclusion criteria was genotype-unrelated injuries or death (single incidents of severe attacks by cage mates, bowel prolapse). No blinding was performed for the analysis of peripheral blood. Spleen volumes were determined by an individual not informed about the cohort assignments. The fulfillment of termination criteria was decided by animal caretakers that were not informed about the cohort assignments.

Cyclophosphamide was injected i.p. at 200 mg/kg once weekly for 4 weeks. Olaparib (Axon Medchem) was dissolved at 50 mg/ml in DMSO, diluted to 5 mg/ml in PBS containing 10% of β-cyclodextrin and injected i.p. five times a week. For the induction of the *Cd19^CreERT2* allele, 4OH-tamoxifen (Sigma-Aldrich) was dissolved in corn oil (Sigma-Aldrich) at 20 mg/ml and injected i.p. at a dose of 120 mg/kg once daily for 5 consecutive days. Palbociclib (LC Laboratories) was dissolved in water at 15 mg/kg and applied by oral gavage at 150 mg/kg once daily for 14 consecutive days.

Animal experiments were conducted with permission of the Landsamt für Natur, Umwelt und Verbraucherschutz Northrhine-Westphalia under the file numbers 84-02.04.2014.A146 and 84-02.04.2014.A083.

**MR imaging**. MR imaging was performed as previously described[22]. In brief, imaging was performed on a clinical 3.0T MRI system (Ingenia, Philips, the Netherlands) with a specific, 40 mm diameter solenoid coil for rodents (Philips Research Europe, Hamburg, Germany) equipped with a heating system to prevent cooling of the anesthetized animals. For anesthesia, 1.5-2.5% isoflurane inhalation was used. The MR images were quantified with the Imalytics research workstation (Philips Innovative Technologies, Aachen, Germany), using semi-automatic segmentation of the spleen.

**Immunoblotting**. $3.5 \times 10^6$ cells were lysed in 100 μl of RIPA buffer and equal volumes were separated on 12.5% SDS-PAGE gels after the addition of Laemmli buffer. Gels were blotted on PVDF membranes (Immobilon-P, Millipore, 0.45 μm). Membranes were stained with specific antibodies against phospho-p53 (1:000, Cell Signaling, Cat. No. 9284), phospho-CHK1 (1:000, Cell Signaling, Cat. No. 2348), and β-actin (1:5000, Sigma, A2228). Proteins were detected using the ECL Western Blotting Detection Kit (GE Healthcare). Uncropped images are given in Supplementary Fig. 11.

**Flow cytometry**. Blood samples were collected from the tail vain. A measure of 20 μl of blood was diluted with 2 ml of ACK buffer and incubated for 3 min at room temperature for erythrocyte lysis. The samples were washed twice with PBS and stained with fluorophore-conjugated antibodies against CD45 (0.5 μl/20 μl blood, APC, Abcam, clone 104-2), CD5 (0.5 μl/20 μl blood, PerCP-Cy5.5, Ebioscience, clone 53-7.3), and CD19 (0.5 μl/20 μl blood, FITC, Ebioscience, clone eBio1D3). Samples were measured on a Beckman Coulter Gallios flow cytometer and analyzed with the software Kaluza (Beckman Coulter). Throughout the manuscript, the size of the CD5$^+$/CD19$^+$ population is given as the percentage of CD45$^+$/SS$^{low}$ lymphocytes.

For total leukocyte and thrombocyte counts and hemoglobin concentration, blood samples were diluted 1:10 with PBS containing 6.08 mM EDTA and analyzed on a XE-5000 (Sysmex).

For the analysis of homing factors surface expression levels, single cells suspensions of leukemic TC, TCP, and TCA mice were prepared and labeled with fluorochrome-conjugated antibodies: PE anti-mouse/human CD44 (clone IM7), FITC anti-mouse CD182 (CXCR2) (clone SA044G4), FITC anti-mouse CD49D (R1-2), PE anti-mouse CD184 (CXCR4) (clone L276F12) from Biolegend; PE-Vio770 anti-mouse CD5, VioBlue anti-mouse CD19, VioGreen anti-mouse CD45 from Miltenyi Biotec, all 1 μl/1E6 cells. Labeled samples were run on a MACSQuant Analyzer VYBe (Miltenyi Biotec), and data were analyzed using MACSQuantify Software (Miltenyi Biotec). Median of fluorescent intensities for CD44, CD49D, CXCR2, and CXCR4 expressions were analyzed on CD5/CD19-positive (leukemic) cells.

**Histology and immunohistochemistry**. Formalin-fixed murine samples were embedded in paraffin and sliced at 2–4 μm. Sections were stained with H/E and antibodies against Ki67 (1:50, Cell Marque) or p21 (undiluted, EuroMabNet, HUGO291). Ki67-positive cells were quantified counting 10 fields of view with a minimum of 200 cells each. p21-positive cells were quantified using the ImmunoRatio software[57].

**Microarray analysis**. RNA was isolated from splenocytes, using guanidinium thiocyanate-phenol-chloroform extraction (Peqgold Trifast, Peqlab). cDNA was synthesized and quality controlled using the Applause WT-Amp Plus ST System (NuGEN), according to the manufacturer's protocol. Mouse Gene ST 2.0 arrays (Affymetrix) were performed by standard protocol and read on a Gene Chip Scanner 3000 7G (Affymetrix).

To identify differentially expressed genes (DEGs), we employed thresholds of $p \leq 0.05$ (Student's paired *t*-test) and an absolute log$_2$ fold change in expression $\geq 0.3$ for each gene (probe set). To interpret the biological function of DEGs, we performed KEGG pathway enrichment analysis based on a hypergeometric test, which was done by the R package GOstats[26].

**Clonality analysis**. For Southern blot analysis of VDJ (Variable, Diversity and Joining gene elements) rearrangement, genomic DNA derived from primary and secondary lymphoid tissues was digested with EcoRI and transferred to a Hybond membrane after gel electrophoresis. The 250 bp probe JH4, derived from the VDJ plasmid via HindIII/NaeI digestions, was labeled with$^{32}$P-α-CTP and results in a 6.2 kb band for germinal center configuration, while VDJ rearranged alleles give bands of different sizes.

For sequence analysis of *Ig* genes, RNA was extracted from lymphocytes isolated from highly infiltrated spleens with TRI Reagent (Sigma, Taufkirchen, Germany). An aliquot of 1 μg RNA was converted to cDNA with QuantiTect Reverse Transcription Kit (Qiagen, Hilden, Germany). Mouse *IgH* rearrangements were analyzed as published previously[58]. Each PCR reaction was analyzed by Qiaxcel Advanced Instrument (Qiagen) by using ScreenGel Software V1.2. Direct sequencing was performed with the ABI Prism Dye Terminator Cycle Sequencing Ready Reaction Kit V3.1 (Life Technologies, Darmstadt, Germany) on an ABI 3130 sequencer (Applied Biosystems, Foster City, CA). Sequences were analyzed using 4Peaks Software V1.7.2 (The Netherlands Cancer Institute, Amsterdam, the Netherlands), Lasergene software (DNAStar, Madison, WI), and manual review. Sequences were compared with mouse germline *Ig* gene sequences with International ImMunoGeneTics database[59].

**Deep sequencing of *Trp53***. For NGS sequencing of the murine *Trp53* locus, DNA was isolated from enlarged spleens of chemotherapy-naive TC mice and animals that had relapsed after four cycles of cyclophosphamide treatment. *Trp53* exons covering the DNA-binding domain (residues 90-300) were amplified by PCR (Phusion Taq polymerase, NEB, primer sequences are listed in Supplementary Table 4). An amount of 100 ng of purified PCR products was used as template for the indexed library preparation. We followed the Illumina TruSeq Nano DNA protocol, skipping the fragmentation step. After library validation and quantification (Agilent 2100 Bioanalyzer), equimolar amounts of library were pooled. Pools of 12 libraries were quantified by using the Peqlab KAPA Library Quantification Kit and the Applied Biosystems 7900HT Sequence Detection System. Pools were sequenced using an Illumina HiSeq 4000 sequencer with a paired-end (76 × 7 × 76 cycles) protocol generating ~700 k reads/sample. The reads were aligned to the mouse GRCm38 reference with the Burrows-Wheeler Aligner (BWA 0.7.15) and the variants were called with the Genome Analysis ToolKit (GATK 3.7 Unified Genotyper) and annotated with Annovar (version 2016-02-01).

**Statistical methods**. Results are presented as mean ± SEM, unless otherwise specified. Cohort sizes for in vivo experiments are noted in the respective figure captions. Cohort sizes were chosen based on experience gathered in previous experiments[22, 60]. For the statistical analysis of survival data, log-rank tests were performed. For the statistical comparison of two independent groups, the unequal variances *t*-test (also known as Welch's *t*-test, an adaption of Student's *t*-test for unequal variances) was used, since variances between groups were generally not

equal (F-test for equal variances). Statistical analyses, where values of the same individuals were compared pre- and post-treatment, were done using a paired t-test. All t-tests were conducted as two-tailed. A specific analysis to test normal distribution of measurements was not performed.

**Human samples**. Human samples were dissolved to a cell suspension. The mononuclear cell fraction was isolated by Ficoll density gradient. DNA was extracted and purified using the Quiagen all prep kit. Quality and quantity of the purified DNA were assessed using the Qubit Fluorometer (Lifetech Technologies) with Qubit dsDNA BR Assay Kit. We used Illumina Design Studio to create custom oligo capture probes flanking each region of interest. For TP53, 15 amplicons with a length of up to 250b covered all coding exons of TP53. Library preparation was performed using TruSeq Custom Amplicon Assay Kit v1.5, including extension and ligation steps between custom probes and adding of indices. Samples were pooled and loaded on a MiSeq flowcell. De-multiplexing, alignment to hg19 reference genome and variant calling was realized via Via MiSeq Reporter pipeline 2.2.9. For annotation and comparison with SNP and COSMIC database, we implemented ensemble variant effect predictor. Local ethics committee approval (EK 138/2003) and patient informed consent was obtained for all experiments conducted with primary human samples in accordance with the Helsinki declaration.

**Data availability**. Microarray data are available through the GEO repository (accession number: GSE98904). All other remaining data are available within the Article and Supplementary Files, or available from the authors on request.

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

## Acknowledgements

We are indebted to our patients, who provided primary material. We thank Alexandra Florin, Marion Müller, and Ursula Rommerscheidt-Fuß from the Institute of Pathology, University Hospital Cologne, for their outstanding technical support. We are grateful to F. Alt (Harvard Medical School) and A. Berns (NKI, Amsterdam) for providing *Atm*$^{fl}$ and *Trp53*$^{fl}$ mice, respectively. This work was supported by the Volkswagenstiftung (Lichtenberg Program, H.C.R.), the Deutsche Forschungsgemeinschaft (KFO-286, H.C.R., M.P., F.T.W., M.H. and SFB1074 subproject B2 to S.S.), the Bundesministerium für Bildung und Forschung (PRECiSe to S.S., 01ZX1406 to M.P. and 01ZX1303A to M.P. and H.C.R.) the Deutsche Jose Carreras Leukämie Stiftung (H.C.R.; R12/08), the Helmholtz-Gemeinschaft (PCCC, H.C.R.), the Else Kröner-Fresenius Stiftung (EKFS-2014-A06, H.C.R.), EC TransCan program (FIRE CLL for S.S.), and the Deutsche Krebshilfe (111724, H.C.R.).

## Author contributions

Conception and design: G.K. and H.C.R. Development of methodology: G.K., T.R., D.K., P.L., A.T., T.P., M.A.-M., and M.M.-R. Acquisition of data (performed experiments, provided animals, acquired and managed patients, provided facilities, etc.): G.K., T.R., D.K., P.L., C.F., A.T., Y.A.-B., M.A.-M., O.F., A.R., P.-H.N., F.T.W., M.M.-R., E.T., S.S., L.F., M.H., C.H., M.H., T.P., R.B. Analysis and interpretation of data (e.g., statistical analysis, biostatistics, and computational analysis): G.K., Y.C., Y.A.-B., F.B., M.M.-R., M.O., E.T., J.B., and M.P. Writing, review, and/or revision of the manuscript: G.K., T.R., and H.C.R. Study supervision: H.C.R.

## Additional information

**Competing interests:** H.C.R. received consulting and lecture fees (Abbvie, Vertex). The remaining authors declare no competing financial interests.

