## [Peer Review File · Nature Communications]

Reviewers' comments:

Reviewer #1 (Remarks to the Author): Expert in CLL mouse models

In this manuscript, the authors addressed the very important and thus far unanswered question how CLL kinetic differ between cases with del17p, del11q/ATM and WT disease. They did so by B cell specific KO of either TP53 or ATM in the Tcl-1 mouse model.

Specific relevant findings are:

- clear differences in survival between the 3 genotypes
- Development of RS in the TP53 genotype
- Detection of specific transcriptomes that surfaces only after induction of genotoxic stress
- confirmation in the mouse model that PARP inhibition specifically benefits ATM deficient mice.

Overall, experiments are very well performed and the data are well presented and scientifically sound.

Specific comments:

Levels most in ATM

Fig 1. The finding that TCA has most rapid increase in CLL percentage in the blood is surprising.

What are the hypothetical mechanisms of this finding?

Fig. 3. To my opinion, the data set of human CLL based on the 2 large trials is somewhat confusing in this paper which focuses on a mouse model. Although of course, it is of interest to compare the outcome of WT vs delATM vs del17p is of relevance, I don't particularly think that it strengthens the mouse data. Specific questions: in what percentage of the human data is known whether the del11q group is heterogeneous with respect to mutation of the remaining ATM locus? Can the authors comment on the fact that in the 2 trials patients with del11q did equally well as WT but according to the data presented here, are doing worse? Is this due to the patients that only had FC or was this also seen in patients with antibody therapy?

Fig 4 and 5 are very well presented and easy to follow; no specific comments

Reviewer #2 (Remarks to the Author): Expert in CLL mouse models

Summary:

The authors present the generation of murine CLL model systems with CD19 specific lack of either ATM or p53 based on the well-used Tcl1 murine model. They show an extensive analysis of the model, fulfilling the expectations of a faithful model of these lesions, allowing for the first time to investigate the impact of these lesions in a tight model. In addition they present a proposed synthetic lethality of ATM mutated CLL in the mouse model and thus define a potential novel avenue of treatment for this subgroup of CLL patients.

The paper is interesting and overall well written, the experiments are well-designed and thoroughly controlled. A few points should be addressed:

Major comments:

1. The authors report that there is no effect of the heterozygous mutations on overall survival.

This is well supported by the data, but may lead to different conclusions.

The important conclusion is that there is no gene dosage effect in this context. But this may not reflect the clinical reality of the CLL patients that are modeled (apart from the option of dominant-acting mutations the outcome of heterozygous lack of ATM or p53 may depend on the selection pressures involved (the gene dosage effect of p53 in the straight p53 deficient mice derives from the selection of loss of heterozygosity in tissues such as thymus where many double strand breaks happen).

It is clear that it would be difficult to test whether cytotoxic treatment of the Tcl1 mice +/-

p53/ATM may select for more aggressive disease experimentally in a meaningful timeframe, but such an experiment would yield important insights into the possible detrimental outcomes associated with cytotoxic agents. As such such a scenario should be discussed in the paper.

2. It is unclear whether the analyses from the CLL8 and CLL10 cohorts (reproducing previously known results in a larger cohort) add much to the current paper or whether it disrupts the flow. But this may be a matter of personal preference.

3. Have the authors checked the DNA damage pathway proficiency of cyclophosphamide treated TC mice by p53 and ATM sequencing or by some biomarker (e.g. challenge (irradiation, Etoposide, etc...)) and transcriptional regulation (p21, Puma, mir34a, etc...)? This may be informative as to what selection (p9, line 31) may have happened.

Minor comments:

1. The second sentence in the abstract (p2, line 3-7 needs some editing).

2. The authors should be more explicit in mentioning the interventional aspect towards the end of the abstract (the word "actionable" alone seems not effective enough to transport the experiments done).

3. The last paragraph of the introduction seems to recapitulate the results more than necessary at this stage (and seems to duplicate parts of results and discussion sections) – this should be shortened

4. the authors should state at what age the olaparib treatment (p12 lines 3ff) was started.

5. the authors should give the genetic background of all their mouse colonies.

6. Figure 1d – the y axis should rather be labeled "platelets" – also the figure may be improved for reading in potential black and white prints by adding different symbols.

Reviewer #3 (Remarks to the Author): Expert in CLL and PARP/DNA damage

In their manuscript entitled "Novel models of high-risk CLL reveal an actionable PARP1 dependence in ATM-defective CLL in vivo" the authors describe two variants of the E μ : TCL driven murine model of CLL (TC), one with the knock out of Trp53 (TCP) and the other with the knock out of Atm (TCA). The author found that both TCP and TCA mice developed CLL at an earlier age and succumbed to disease earlier than control TC mice. They subsequently compared global transcription profiles of the splenic cells in these murine models to TC splenic cells and observed that in addition to apoptotic defects, TCP mice showed over-representation of cell cycle genes, whereas TCA mice also exhibited cell cycle gene deregulation but retained apoptotic transcriptional response. All three models were equally sensitive to chemotherapy, but response was transient. Finally, based on the observation that PARP was overexpressed in TCA mice, the authors tested this CLL model for sensitivity to PARP inhibition and observed that TCA CLLs were differentially sensitive to this treatment.

This is an interesting, well written paper that describes two important new tools (models) to address biology of aggressive CLL associated with ATM or p53 dysfunction. However, the paper does not offer new concepts or insights and merely demonstrates what is already known in the CLL field.

My specific concepts are as follows:

-The authors draw parallels between CLL8 and CLL10 trials, comparing patients with 11q or 17p deletion with TCA and TCP murine models. These comparisons are not valid, as the status of the second allele is not known in CLL10 and CLL8 patients. CLLs with 11q and 17p deletion in these trials refer to monoallelic loss of ATM or TP53 respectively, whereas TCA and TCP models refer to biallelic loss of the same genes. Indeed, their results showing that heterozygous Trp53 or Atm mice do not have reduced survival compared to the TC control mice, also argues against the validity of the above comparison.

- In their treatment experiments the authors assess murine peripheral blood CLL cells as a reflection of tumour load. This is not a valid approach as tumour cells populate the spleens (particularly in the TCP model) and therefore, the counting of PB CLL cells only is not a valid measurement of response to treatment. The authors should address both PB tumour load and splenic CLL infiltration at fixed time points following treatment. Indeed, in Figure 1c the authors show that CLL counts in PB are highest in TCA mice and yet TCP mice survive for shorter time. This also argues in support of the notion that there is significant tumour load away from the PB.

-In CLL patients ATM and TP53 transcriptional pro-apoptotic pathways overlap. What is the explanation for the fact that in TCA model TP53 pro-apoptotic transcription is working? The authors suggest the role of an alternative kinase, but they should provide more evidence. The age of the animals used for the microarrays is also important and not clear. Were spleens on analysed animals already affected by initial CLL infiltration at the time of analysis? This information is important as it could affect differences in the transcription profiles.

-Based on the transcriptional PARP upregulation the authors treat TCA tumours with PARP inhibitors, what about TCP tumours? Would cdk inhibitors be effective in TCP mice giving differential upregulation of cyclin genes in this model?

Reviewer #1

In this manuscript, the authors addressed the very important and thus far unanswered question how CLL kinetic differ between cases with del17p, del11q/ATM and WT disease. They did so by B cell specific KO of either TP53 or ATM in the Tcl-1 mouse model.

Specific relevant findings are:

- *clear differences in survival between the 3 genotypes*
 - *Development of RS in the TP53 genotype*
 - *Detection of specific transcriptomes that surfaces only after induction of genotoxic stress*
 - *confirmation in the mouse model that PARP inhibition specifically benefits ATM deficient mice.*
- Overall, experiments are very well performed and the data are well presented and scientifically sound.*

Thank you for this kind assessment of our work.

Specific comments:

Fig 1. The finding that TCA has most rapid increase in CLL percentage in the blood is surprising. What are the hypothetical mechanisms of this finding?

Thank you for raising this important point. We did go back to the original data and asked whether the percentage of Cd5⁺/Cd19⁺ cells differed between TCP and TCA animals at any of the time points indicated in **Fig. 1c**. We note that the percentages do not significantly differ between these two genotypes at any time point.

However, as noted by this reviewer, our data indicate a trend towards an increased percentage of Cd5⁺/Cd19⁺ cells in TCA versus TCP mice. One potential mechanism may be a difference in the surface expression of homing factors between the malignant cells of the different genotypes. Although not specifically asked by this reviewer, we directly addressed this possibility by performing flow cytometry experiments to assess CXCR4, CXCR2, CD44 and CD49D surface expression on the leukemic cells, as all of these factors were suggested to mediate CLL homing and growth-supporting contact to the microenvironment¹⁻⁵. As shown in the new **Suppl. Fig. S1**, we did not observe any significant differences in the surface expression of these homing factors, when comparing TC, TCA and TCP leukemia cells. Thus, overall, we note that the percentage of leukemic cells does not significantly differ between TCA and TCP mice, although there is a trend towards higher peripheral cell counts in TCA, compared to TCP mice. As best as we can tell from our new data, the surface expression of CXCR4, CXCR2, CD44 and CD49D does not explain this trend. The new text passage reads as follows:

... At 12 weeks of age, 9/20 of TCA and 12/20 TCP mice had a clearly identifiable leukemic population (Cd5⁺/Cd19⁺ ≥ 5%), compared to only 3/20 of TC controls (p = 0.004 for TC vs. TCP and p = 0.041 for TC vs. TCA, Fisher's exact test). Somewhat surprisingly, there was a trend, albeit not significant, towards an earlier occurrence of a leukemic population in the peripheral blood of TCA, compared to TCP animals (**Fig. 1c**). One potential mechanism for this observation may be a difference in the surface expression of homing factors between the malignant cells of the different genotypes. We addressed this possibility, by performing flow cytometry experiments to assess CXCR4, CXCR2, CD44 and CD49D surface expression on the leukemic cells, as all of these factors were suggested to mediate CLL homing and growth-supporting contact to the microenvironment³²⁻³⁶. However, we did not observe any significant differences in the surface expression of these factors, when comparing TCA and TCP leukemia cells (**Fig. S1**). Consistent with enhanced leukemogenesis and subsequent bone marrow failure, we observed significantly lower platelet counts in TCP and TCA animals, compared to TC controls, at 16 weeks of age (p ≤ 0.021, **Fig. 1d**). ...

Fig. 3. To my opinion, the data set of human CLL based on the 2 large trials is somewhat confusing in this paper which focuses on a mouse model. Although of course, it is of interest to compare the outcome of WT vs delATM vs del17p is of relevance, I don't particularly think that it strengthen the mouse data. Specific questions: in what percentage of the human data is known whether the del11q group is heterogeneous with respect to mutation of the remaining ATM locus? Can the authors

comment on the fact that in the 2 trials patients with del11q did equally well as WT but according to the data presented here, are doing worse? Is this due to the patients that only had FC or was this also seen in patients with antibody therapy?

Thank you very much for raising this point. In fact, all three reviewers felt that the discussion of the human data might be confusing or distracting in this manuscript. In re-reading the paper, we fully agree with the three independent reviewers and hence removed the human data from this paper to strictly focus on the novel mouse models and their biology.

With regard to the specific questions raised by this reviewer: We do not have ATM DNA sequencing data on the majority of the del(11q) patients. With regard to the second point: Yes, this is due to the patients that did not receive rituximab. Furthermore, our animals only received cyclophosphamide, rather than cyclophosphamide/fludarabine or cyclophosphamide/fludarabine/rituximab. Given all these differences between our mouse system and the human data sets, we felt comfortable in removing the patient data from the current manuscript.

Fig 4 and 5 are very well presented and easy to follow; no specific comments

Thank you very much for this kind assessment.

Reviewer #2 (Remarks to the Author): Expert in CLL mouse models

The authors present the generation of murine CLL model systems with CD19 specific lack of either ATM or p53 based on the well-used *Tcl1* murine model. They show an extensive analysis of the model, fulfilling the expectations of a faithful model of these lesions, allowing for the first time to investigate the impact of these lesions in a tight model. In addition they present a proposed synthetic lethality of ATM mutated CLL in the mouse model and thus define a potential novel avenue of treatment for this subgroup of CLL patients.

The paper is interesting and overall well written, the experiments are well-designed and thoroughly controlled. A few points should be addressed:

Thank you very much for this kind assessment of our work.

Major comments:

1. The authors report that there is no effect of the heterozygous mutations on overall survival. This is well supported by the data, but may lead to different conclusions.

The important conclusion is that there is no gene dosage effect in this context. But this may not reflect the clinical reality of the CLL patients that are modeled (apart from the option of dominant-acting mutations the outcome of heterozygous lack of ATM or p53 may depend on the selection pressures involved (the gene dosage effect of p53 in the straight p53 deficient mice derives from the selection of loss of heterozygosity in tissues such as thymus where many double strand breaks happen).

It is clear that it would be difficult to test whether cytotoxic treatment of the *Tcl1* mice +/- p53/ATM may select for more aggressive disease experimentally in a meaningful timeframe, but such an experiment would yield important insights into the possible detrimental outcomes associated with cytotoxic agents. As such such a scenario should be discussed in the paper.

Thank you for raising this point. Unfortunately, we did not have heterozygous animals available at the time that this review was written. Given the extensive breeding time and the extraordinary latency of our models, it simply is impossible to perform these experiments within the time frame of this revision process, which is approximately 6 months. We are grateful that this reviewer appreciates the difficulty of achieving these experiments in a "meaningful" timeframe. However, we feel that the points raised by this reviewer are important and should probably be addressed in future work. Following this reviewer's advice, we included a passage into our discussion section that is devoted to this topic. The relevant passage now reads as follows:

... Most importantly, and in full agreement with the human situation, both TCP and TCA animals succumb to their disease significantly earlier, than their standard-risk TC counterparts (**Fig. 1g**). Our data further indicate that heterozygous deletion of *Trp53* or *Atm* does not *per se* lead to a significant reduction in overall survival, compared to the *Em:TCL1;Cd19^{Cre/wt}* parental model (**Fig. S4**). However, we note that we have not challenged the heterozygotes with genotoxic therapy. These experiments may reveal important mechanisms of chemotherapy-induced selection of aggressive clones, which could be mediated by loss of heterozygosity or the acquisition of protein-damaging mutations on the remaining *Trp53* and *Atm* alleles. Thus, these experiments should be performed in the future. ...

2. It is unclear whether the analyses from the CLL8 and CLL10 cohorts (reproducing previously known results in a larger cohort) add much to the current paper or whether it disrupts the flow. But this may be a matter of personal preference.

Thank you very much for raising this point. As detailed in our response to reviewer #1, all three reviewers felt that the discussion of the human data might be confusing or distracting in this manuscript. In re-reading the paper, we fully agree with the reviewing experts and hence removed the human data from this paper to strictly focus on the novel mouse models and their biology.

3. Have the authors checked the DNA damage pathway proficiency of cyclophosphamide treated TC mice by p53 and ATM sequencing or by some biomarker (e.g. challenge (irradiation, Etoposide, etc...) and transcriptional regulation (p21, Puma, mir34a, etc...)? This may be informative as to what selection (p9, line 31) may have happened.

Thank you very much for raising this point. We performed extensive *in vivo* experiments to address this important point. We specifically assessed the functionality of the DNA damage response in TC animals that were chemotherapy-naïve and in TC animals that had relapsed after 4 cycles of cyclophosphamide. We assessed the DNA damage response by performing immunohistochemistry to detect the accumulation of the p53 target gene p21 following acute challenge with cyclophosphamide. As a control, we included leukemic TCP animals, as p21 should not be induced in these animals due to a lack of p53. In addition, we performed deep genomic sequencing of *Trp53* in CLL samples of chemotherapy-naïve leukemic TC animals, as well as TC animals that had relapsed following four cycles of cyclophosphamide.

We essentially found that p21 was robustly induced in leukemic spleens of chemotherapy-naïve TC animals following acute challenge with cyclophosphamide, whereas p21 was induced to a significantly lower extent in leukemic spleens of TC animals that were challenged with cyclophosphamide following the development of CLL relapse after 4 cycles of cyclophosphamide. This data is now included as **Suppl. Figure S8a and b**. This impaired induction of the p53 response in leukemia-infiltrated spleens of TC mice is also mirrored by an impaired Ser-18 phosphorylation in p53 in these animals. While p53 was uniformly and robustly phosphorylated on Ser-18 in CLL-infiltrated spleens of acutely cyclophosphamide-challenged, but previously chemotherapy-naïve TC mice, this phosphorylation was not detectable in all leukemia-infiltrated spleens of TC mice that had relapsed following four cycles of cyclophosphamide (**Suppl. Figure S8c and d**). Following this reviewer's advise, we also sequenced the *Trp53* gene in CLL-infiltrated spleens of chemotherapy-naïve TC mice and in TC animals that had relapsed after 4 cycles of cyclophosphamide. Somewhat surprisingly we did not detect any mutations within the cDNA encoding the DNA binding domain, which contains the hotspot areas for mutations. Together, these observations suggest that the p53 pathway is altered in relapsed animals through mechanisms that are not dominated by protein-damaging *Trp53* mutations. The relevant new text passage now read as follows:

... These experiments revealed that CLL progressed with significantly reduced kinetics in TC, compared to both TCA ($p = 0.038$) and TCP ($p = 0.007$) animals prior to cyclophosphamide exposure, mirroring the less aggressive disease characteristics in TC animals (**Fig. 3g, h, S7a-c**). In marked contrast, leukemia progression was drastically enhanced in TC animals that had relapsed following cyclophosphamide treatment. This post-treatment progression rate was not statistically different between TC, TCA and TCP animals ($p \geq 0.41$, **Fig. 3h**), strongly suggesting that four cycles of cyclophosphamide were insufficient to cure the disease in TC animals and instead selected a clone(s) that displayed similar aggressiveness as those developing in TCA and TCP animals. To further investigate the functionality of the p53 pathway in CLL cells of TC animals that had relapsed following 4 cycles of cyclophosphamide, we performed immunohistochemistry to assess the induction of the *bona fide* p53 target gene p21. These experiments revealed that exposure of chemotherapy-naïve leukemic TC animals led to a substantial induction of p21 in the spleen-infiltrating leukemic cells (**Fig. S8a, b**). In marked contrast, p21 was only marginally induced in spleen-infiltrating CLL cells of TC animals that were re-challenged with cyclophosphamide, following relapse after 4 cycles of cyclophosphamide (**Fig. S8a, b**). In line with these data, we found that p53 Ser-18 phosphorylation was strongly induced in spleen lysates of leukemic TC animals upon cyclophosphamide treatment (**Fig. S8c**). In contrast, p53 Ser-18 phosphorylation was induced by cyclophosphamide re-challenge only in approx. 50% of spleen lysates derived from TC animals that had relapsed following 4 cycles of cyclophosphamide (**Fig. S8d**). Lastly, we assessed the mutation status of *Trp53* in CLL-infiltrated spleens of chemotherapy-naïve TC mice and in TC animals that had relapsed after 4 cycles of cyclophosphamide. For that purpose we designed primers to amplify the regions coding for the DNA-binding domain, which harbors most mutation hot spots found in human cancers⁴⁶. However, after deep sequencing of the amplicons at a minimum coverage of more than 20,000 X

(Table S1), we did not detect any mutation in either the primary or the relapsed CLL. Together, these observations suggest that the p53 pathway is altered in relapsed animals through mechanisms that are not dominated by protein-damaging *Trp53* mutations. These *in vivo* observations are corroborated by clinical data from our patient data-base. ...

Minor comments:

1. *The second sentence in the abstract (p2, line 3-7 needs some editing).*

Thank you for picking this up. We have shortened and adapted the sentence to now read as follows:

... Two cytogenetic aberrations, namely del(17p), affecting TP53, and del(11q), affecting ATM, are associated with inherent chemotherapy-resistance (del17p) and poor outcome (del11q, del17p). ...

2. *The authors should be more explicit in mentioning the interventional aspect towards the end of the abstract (the word "actionable" alone seems not effective enough to transport the experiments done).*

Thank you for picking this up. We have adapted the sentence to now read as follows:

... We further employ these novel models in conjunction with transcriptome analyses following cyclophosphamide treatment to reveal that *Atm*-deficiency is associated with an exquisite and genotype-specific sensitivity against PARP inhibition. ...

3. *The last paragraph of the introduction seems to recapitulate the results more than necessary at this stage (and seems to duplicate parts of results and discussion sections) – this should be shortened*

We fully agree with this reviewer and have thus shortened the last part of the introduction. This part now reads as follows:

... Here, we generated and characterized *Eμ:TCL1*-driven autochthonous models of high-risk CLL that are characterized by conditional B cell-specific deletion of *Atm* or *Trp53*. Our novel models of *Atm*- or *Trp53*-deficient CLL might serve as ideal preclinical platforms for the discovery and *in vivo* validation of molecular liabilities associated with these high-risk genetic aberrations in CLL. ...

4. *the authors should state at what age the olaparib treatment (p12 lines 3ff) was started.*

Thank you for raising this point. We apologize for being unclear on this point. We have now specified the treatment schedule. The relevant passage now reads:

... We thus exposed leukemic (defined as $\geq 15\%$ of all CD45⁺/SS^{low} cells displaying the CD5⁺/CD19⁺ immunophenotype of CLL cells and/or having a spleen volume of $\geq 200\mu\text{l}$) TC (median age of 31.8 \pm 4.0 weeks at treatment initiation) and TCA mice (median age of 23.6 \pm 6.1 weeks at treatment initiation) to olaparib (50 mg/kg, intraperitoneally on 5 days per week) and assessed leukemia progression through flow cytometry-based quantification of CD5⁺/CD19⁺ leukemic cells in the blood stream and MRI-based quantification of spleen volume, as a surrogate marker for lymphadenopathy. ...

5. *the authors should give the genetic background of all their mouse colonies.*

Thank you for raising this important point. We apologize for being unclear on this. All of our mouse models were generated on a mixed C57BL/6J-C57BL/6N background. We have added this information to the Materials and Methods section. The relevant part now reads as follows:

... B cell-specific loss of *Atm* and *Trp53* in *Eμ:TCL1* mice was modeled by crossing the *Eμ:TCL1* allele with *Cd19^{Cre}* and either *LoxP*-flanked *Atm* or *Trp53* alleles on a mixed C57BL/6J-C57BL/6N background. For the tamoxifen-inducible activation of Cre recombinase, the *Cd19^{CreERT2}* allele was used in a hemizygous state on a mixed C57BL/6J-C57BL/6N background. ...

6. Figure 1d – the y axis should rather be labeled “platelets” – also the figure may be improved for reading in potential black and white prints by adding different symbols.

We agree with this reviewer. The axis labeling has been modified accordingly. We also used distinct symbols for the different genotypes to enhance black and white readability.

Reviewer #3

In their manuscript entitled "Novel models of high-risk CLL reveal an actionable PARP1 dependence in ATM-defective CLL in vivo" the authors describe two variants of the Eμ: TCL driven murine model of CLL (TC), one with the knock out of Trp53 (TCP) and the other with the knock out of Atm (TCA). The author found that both TCP and TCA mice developed CLL at an earlier age and succumbed to disease earlier than control TC mice. They subsequently compared global transcription profiles of the splenic cells in these murine models to TC splenic cells and observed that in addition to apoptotic defects, TCP mice showed over-representation of cell cycle genes, whereas TCA mice also exhibited cell cycle gene deregulation but retained apoptotic transcriptional response. All three models were equally sensitive to chemotherapy, but response was transient. Finally, based on the observation that PARP was overexpressed in TCA mice, the authors tested this CLL model for sensitivity to PARP inhibition and observed that TCA CLLs was differentially sensitive to this treatment.

This is an interesting, well written paper that describes two important new tools (models) to address biology of aggressive CLL associated with ATM or p53 dysfunction. However, the paper does not offer new concepts or insights and merely demonstrates what is already known in the CLL field.

Thank you very much for this kind assessment of our work.

My specific concepts are as follows:

-The authors draw parallels between CLL8 and CLL10 trials, comparing patients with 11q or 17p deletion with TCA and TCP murine models. These comparisons are not valid, as the status of the second allele is not known in CLL10 and CLL8 patients. CLLs with 11q and 17p deletion in these trials refer to monoallelic loss of ATM or TP53 respectively, whereas TCA and TCP models refer to biallelic loss of the same genes. Indeed, their results showing that heterozygous Trp53 or Atm mice do not have reduced survival compared to the TC control mice, also argues against the validity of the above comparison.

Thank you very much for raising this point. As detailed in our response to reviewers #1 and #2, all three reviewers felt that the discussion of the human data might be confusing or distracting in this manuscript. In re-reading the paper, we fully agree with the reviewing experts and hence removed the human data from this paper to strictly focus on the novel mouse models and their biology.

- In their treatment experiments the authors assess murine peripheral blood CLL cells as a reflection of tumour load. This is not a valid approach as tumour cells populate the spleens (particularly in the TCP model) and therefore, the counting of PB CLL cells only is not a valid measurement of response to treatment. The authors should address both PB tumour load and splenic CLL infiltration at fixed time points following treatment. Indeed, in Figure 1c the authors show that CLL counts in PB are highest in TCA mice and yet TCP mice survive for shorter time. This also argues in support of the notion that there is significant tumour load away from the PB.

Thank you for raising this important issue. We fully agree that an assessment of both peripheral blood counts and splenomegaly should be assessed to fully capture therapy response in our novel models. Following this reviewer's suggestion, we have now assessed the percentage of peripheral CD5⁺/CD19⁺ cells, peripheral leukocyte counts, platelet counts and spleen volume (measured by MR imaging) prior to and following 4 cycles of cyclophosphamide. All measurements were taken 7 days following the last cyclophosphamide dose. These experiments clearly revealed that cyclophosphamide does not only lead to a substantial reduction of leukemic cells in the peripheral blood, but that cyclophosphamide treatment also led to a substantial reduction in splenomegaly. Overall, we feel that this dual assessment of cyclophosphamide response has enhanced the quality and validity of our data and we are thankful to this reviewer for raising this point. The response assessment data are now included as **Figure 3a-e**. The relevant text passage now reads as follows:

... Acute response to therapy was assessed 7 days following the last cyclophosphamide

dose through flow cytometry-based quantification of the leukemic clone in the peripheral blood. As shown in **Fig. 3a** and **b**, chemotherapy treatment led to a significant reduction of the overall leukocyte count and the leukemic burden (CD5⁺/CD19⁺ cells) in all three genotypes. Paralleling this suppression of the leukemic cells, we observed a recovery of the platelet count in the blood stream, which we interpret as a sign of bone marrow recovery (**Fig. 3c**). This hematological response was also paralleled by MRI-morphological regression of splenomegaly in the cyclophosphamide-treated animals. As shown in **Fig. 3d** and **e**, MRI-based spleen volumetry demonstrated a substantial reduction in spleen volume in all three genotypes (TC, TCA, TCP), 7 days following the last cyclophosphamide dose. ...

-In CLL patients ATM and TP53 transcriptional pro-apoptotic pathways overlap. What is the explanation for the fact that in TCA model TP53 pro-apoptotic transcription is working? The authors suggest the role of an alternative kinase, but they should provide more evidence. The age of the animals used for the microarrays is also important and not clear. Were spleens on analysed animals already affected by initial CLL infiltration at the time of analysis? This information is important as it could affect differences in the transcription profiles.

Thank you for pointing out these important questions. We have performed immunoblotting to assess the activity of the ATR/CHK1 axis in TCA animals following acute *in vivo* challenge with cyclophosphamide. We particularly blotted pCHK1 (pSer-345, the Ser-Gln motif that is phosphorylated by the proximal DNA damage response kinase ATR) pp53 (pSer-18, the Ser-Gln that is phosphorylated by the proximal DNA damage response kinases ATR, DNA-PKcs and ATM). These experiments demonstrated that both CHK1 and p53 are phosphorylated in a cyclophosphamide-dependent manner in TCA animals. Particularly the CHK1 pSer-345 epitope has been shown to be generated in an ATR-dependent manner and thus indicates that the ATR/CHK1 pathway is intact in the TCA animals⁶⁻⁹. Furthermore, the p53 Ser-18 site has been demonstrated to be the substrate site of all three proximal DNA damage response kinases (ATM, ATR, DNA-PKcs)^{6,7,10-13}. The fact that this site is phosphorylated in the absence of ATM in our TCA animals strongly argues that ATR and/or DNA-PKcs are activated in TCA animals following cyclophosphamide challenge. These data are now included as **Figure S9**. The relevant section now reads as follows:

... However, activation of *Trp53*-mediated apoptotic signaling still occurs, most likely via other proximal DNA damage response kinases sharing substrate homology with ATM, such as ATR or DNA-PKcs (illustrated in **Fig. 4d**). In line with this hypothesis, we found that CHK1 is phosphorylated on the ATR substrate site Ser³⁴⁵-Gln³⁴⁶ in lysates generated from leukemia-infiltrated spleens isolated from TCA animals 12 hours following cyclophosphamide challenge, *in vivo* (**Fig. S9**). The Ser-345 residue has previously been shown to be phosphorylated in an ATR-dependent manner^{6,47-49}. Moreover, we also observed robust phosphorylation of Ser-18 in p53 in these lysates (**Fig. S9**). The Ser-18 site is a well-established substrate of the Ser-Gln-directed proximal DNA damage response kinases ATM, ATR and DNA-PKcs^{6,47,50-54}. Thus, overall, these immunoblot experiments strongly indicate that ATR and/or DNA-PKcs remain active in TCA CLL cells. ...

The age of the animals was 42.8±5.4 weeks (TC), 34.6±5.2 weeks (TCP) and 31.8±7.5 weeks (TCA) at the time of tissue isolation for microarray analysis. All of the animals included in the microarray experiments displayed MRI-morphologically verified enlarged spleens at the time of sacrifice. We have also clarified these issues in the main text. The relevant section now reads as follows:

... To further characterize the chemotherapy response in the TC, TCA and TCP models, we next performed microarray-based gene expression analysis on splenocytes derived from leukemic animals with MRI-morphologically verified splenomegaly. At the time of sacrifice, TC animals were 42.8±5.4, TCA animals were 31.8±7.5 and TCP mice were 34.6±5.2 weeks of age. ...

-Based on the transcriptional PARP upregulation the authors treat TCA tumours with PARP inhibitors, what about TCP tumours? Would cdk inhibitors be effective in TCP mice giving differential upregulation of cyclin genes in this model?

Thank you for this suggestion. We note that *Ccne* is upregulated in TCP animals following cyclophosphamide treatment (**Figure 4b**, right panel). In contrast, *Parp* expression was relatively increased in CLL-infiltrated spleens of untreated TCA animals, compared to CLL-infiltrated spleens of untreated TC animals. Thus, there is a fundamental difference between the cyclophosphamide-induced increased *Ccne* expression in TCP animals and the relatively increased *Parp* expression in untreated TCA vs. TC animals. Nevertheless, we performed an extensive set of *in vivo* experiments to assess a potential therapeutic efficacy of the Cdk4/6 inhibitor palbociclib in TC and TCP animals. These experiments revealed that palbociclib did induce a mild reduction in splenomegaly in both leukemic TC and TCP animals. However, these effects did not reach statistical significance and were observed in TC and TCP animals, arguing against a genotype-specific effect. In addition, we did observe a mild palbociclib-induced reduction in leukemic cells in the peripheral blood of TC and TCP animals. However, the differences between the two genotypes were not significant. Overall, our data indicate that the Cdk4/6 inhibitor palbociclib displays a mild, but not TCP-specific, anti-leukemic effect in our CLL mouse models. These new data are now included as **Suppl. Figure S10**. The relevant new text passage now reads as follows:

... One of the most striking observations within our transcriptome analyses was the upregulation of the G₁/S cyclin gene *Ccne* in TCP animals following cyclophosphamide (**Fig. 4b**). These data could suggest that Cdk4/6 inhibition may be particularly effective in TCP animals. To directly address this possibility, we performed a set of *in vivo* experiments to assess a potential therapeutic efficacy of the Cdk4/6 inhibitor palbociclib in TC and TCP animals. These experiments revealed that palbociclib did induce a mild reduction in splenomegaly in both leukemic TC and TCP animals (**Fig. S10a, b**). However, these effects did not reach statistical significance and were observed both, in TC and TCP animals, arguing against a genotype-specific effect (**Fig. S10a, b**). In addition, we did observe a mild palbociclib-induced reduction in leukemic cells in the peripheral blood of TC and TCP animals (**Fig. S10c**). However, the differences between the two genotypes were not significant. Overall, these data indicate that palbociclib displays a mild, but not TCP-specific, anti-leukemic effect in our CLL mouse models. ...

References

- 1 Burger, J. A., Burger, M. & Kipps, T. J. Chronic lymphocytic leukemia B cells express functional CXCR4 chemokine receptors that mediate spontaneous migration beneath bone marrow stromal cells. *Blood* **94**, 3658-3667 (1999).
- 2 Levidou, G. *et al.* Immunohistochemical analysis of IL-6, IL-8/CXCR2 axis, Tyr p-STAT-3, and SOCS-3 in lymph nodes from patients with chronic lymphocytic leukemia: correlation between microvascular characteristics and prognostic significance. *BioMed research international* **2014**, 251479, doi:10.1155/2014/251479 (2014).
- 3 Fedorchenko, O. *et al.* CD44 regulates the apoptotic response and promotes disease development in chronic lymphocytic leukemia. *Blood* **121**, 4126-4136, doi:10.1182/blood-2012-11-466250 (2013).
- 4 Dal Bo, M. *et al.* Microenvironmental interactions in chronic lymphocytic leukemia: the master role of CD49d. *Semin Hematol* **51**, 168-176, doi:10.1053/j.seminhematol.2014.05.002 (2014).
- 5 Till, K. J., Lin, K., Zuzel, M. & Cawley, J. C. The chemokine receptor CCR7 and alpha4 integrin are important for migration of chronic lymphocytic leukemia cells into lymph nodes. *Blood* **99**, 2977-2984 (2002).
- 6 Reinhardt, H. C. & Yaffe, M. B. Kinases that control the cell cycle in response to DNA damage: Chk1, Chk2, and MK2. *Current opinion in cell biology* **21**, 245-255, doi:10.1016/j.ceb.2009.01.018 (2009).
- 7 Reinhardt, H. C. & Yaffe, M. B. Phospho-Ser/Thr-binding domains: navigating the cell cycle and DNA damage response. *Nat Rev Mol Cell Biol* **14**, 563-580, doi:10.1038/nrm3640 (2013).
- 8 Liu, Q. *et al.* Chk1 is an essential kinase that is regulated by Atr and required for the G(2)/M DNA damage checkpoint. *Genes Dev* **14**, 1448-1459 (2000).
- 9 Zhao, H. & Piwnicka-Worms, H. ATR-mediated checkpoint pathways regulate phosphorylation and activation of human Chk1. *Mol Cell Biol* **21**, 4129-4139, doi:10.1128/MCB.21.13.4129-4139.2001 (2001).
- 10 Banin, S. *et al.* Enhanced phosphorylation of p53 by ATM in response to DNA damage. *Science* **281**, 1674-1677 (1998).
- 11 Canman, C. E. *et al.* Activation of the ATM kinase by ionizing radiation and phosphorylation of p53. *Science* **281**, 1677-1679 (1998).
- 12 Tibbetts, R. S. *et al.* A role for ATR in the DNA damage-induced phosphorylation of p53. *Genes Dev* **13**, 152-157 (1999).
- 13 Woo, R. A., McLure, K. G., Lees-Miller, S. P., Rancourt, D. E. & Lee, P. W. DNA-dependent protein kinase acts upstream of p53 in response to DNA damage. *Nature* **394**, 700-704, doi:10.1038/29343 (1998).

REVIEWERS' COMMENTS:

Reviewer #1 (Remarks to the Author):

The authors have addressed all my raised question in a satisfying manner. I have no further comments and advise accepting this manuscript.

Reviewer #2 (Remarks to the Author):

The authors have made reasonable efforts to answer all questions raised and improved the manuscript in content and interpretation. I have no more improvements of scientific content to ask.

One issue of style might be addressed. Page 10 lines 7pp relates to patient materials. Since most of the clinical dataset has been removed, this feels somewhat insular and might need a specific short introduction. Also there are no longer any parts of the methods section addressing the human studies. This should be corrected. As an alternative the data might be removed – relevant datasets with regards to selection of p53 mutated clones exist by Rossi et al, thus the novelty of the observation is limited and thus the presentation is not absolutely necessary.

Reviewer #3 (Remarks to the Author):

The authors addressed all my comments. I have not anything further to add.

Reviewer comments:

Reviewer #2:

The authors have made reasonable efforts to answer all questions raised and improved the manuscript in content and interpretation. I have no more improvements of scientific content to ask.

Thank you very much for your help in improving the manuscript.

One issue of style might be addressed. Page 10 lines 7pp relates to patient materials. Since most of the clinical dataset has been removed, this feels somewhat insular and might need a specific short introduction.

We agree with this reviewer and have hence incorporated a very short introductory section, which reads as follows:

... These *in vivo* observations are corroborated by clinical data from our patient data-base. Specifically, we identified two individual patients that received fludarabine/cyclophosphamide-based chemotherapy and monitored the occurrence and clonal dynamics of CLL-associated high-risk mutations, using a targeted next generation sequencing approach. ...

Also there are no longer any parts of the methods section addressing the human studies. This should be corrected.

We agree with this reviewer and have modified the Materials and Methods section accordingly. The relevant part now reads as follows:

... *Human samples*

Human samples were dissolved to a cell suspension. The mononuclear cell fraction was isolated by Ficoll density gradient. DNA was extracted and purified using the Quiagen all prep kit. Quality and quantity of the purified DNA were assessed using the Qubit Fluorometer (Lifetech technologies) with Qubit dsDNA BR Assay Kit. We used Illumina Design Studio to create custom oligo capture probes flanking each region of interest. For *TP53*, 15 amplicons with a length of up to 250b covered all coding exons of *TP53*. Library preparation was performed using TruSeq Custom Amplicon Assay Kit v1.5 including extension and ligation steps between custom probes and adding of indices. Samples were pooled and loaded on a MiSeq flowcell. De-multiplexing, alignment to hg19 reference genome and variant calling was realized via Via MiSeq Reporter pipeline 2.2.9. For Annotation and Comparison with SNP- and COSMIC Database, we implemented Ensemble variant effect predictor. Local ethics committee approval (EK 138/2003) and patient informed consent was obtained for all experiments conducted with primary human samples in accordance with the Helsinki declaration. ...

We hope that these responses clarified all the points raised by Reviewer #2.